# Technical note: Bimodal Parameterizations of in situ Ice Cloud Particle Size Distributions

Irene Bartolomé García[1,a], Odran Sourdeval[2], Reinhold Spang[1], and Martina Krämer[1,3]

[1]Institute for Energy and Climate Research (IEK-7), Research Center Jülich, D-52425 Jülich, Germany
[2]University of Lille, CNRS, UMR 8518-LOA-Laboratoire d'Optique Atmosphérique, F-59000 Lille, France
[3]Institute for Physics of the Atmosphere (IPA), Johannes-Gutenberg University, D-55122 Mainz, Germany
[a]Now at: Institute of Geophysics and Meteorology, University of Cologne, Cologne, Germany

**Correspondence:** ibartolo@uni-koeln.de, m.kraemer@fz-juelich.de

**Abstract.** The cloud particle size distribution (PSD) is a key parameter for the retrieval of micro-physical and optical properties from remote sensing instruments, which in turn are necessary for determining the radiative effect of clouds. Current representations of PSDs for ice clouds rely on parameterizations that were largely based on aircraft in situ measurements where the distribution of small ice crystals were uncertain. This makes current parameterizations deficient to simulate remote sensing observations sensitive to small ice, such as from lidar and thermal infrared instruments. In this study we fit the in situ PSDs of ice crystals from the JULIA (JÜLich In situ Aircraft data set) database, which consists of 11 campaigns covering the tropics, mid-latitudes and the Arctic, consistently processes and considered more robust in their measurements of small ice. For the fitting, we implement an established approach to PSD parameterizations, which consists of finding an adequate set of parameters for a modified gamma function after normalization of both PSD axes. These parameters are constrained to match in-situ measurements when predicting microphysical properties from the PSDs, via a cost function minimization method. We selected the ice water content and the ice crystal number concentration, which are currently key parameters for modern satellite retrievals and model microphysics schemes. We found that a bimodal parameterization yields better results than a monomodal one. The bimodal parameterization has a lower spread for almost all ice crystal sizes over the entire range of analyzed temperatures and fits better the observations, especially for particles between 20 and about $110\,\mu$m at temperatures between $-60$ and $-20\,°$C. For this temperature range, the root mean square error for the retrieved $N_{ice}$ is reduced from 0.36 to 0.20. This demonstrates a clear advantage to considering the bimodality of PSDs, e.g. for satellite retrievals.

## 1 Introduction

Ice clouds play an important role in Earth's radiative budget, as their radiative effects can either contribute to a warming or a cooling of the surface (Liou, 1986; Stephens et al., 1990). The balance between these two effects depends on their macro- and micro-physical properties, which stem from an array of very complex processes governing ice crystal formation and growth (Zhang et al., 1999; Krämer et al., 2020). Despite extensive research over the past decades most of these processes remain highly uncertain, making ice clouds major unknowns in current climate studies (Bellouin et al., 2020). Ice clouds are particularly challenging for satellite remote-sensing techniques, in great part due to the complexity and variety of their

microphysical and optical properties (Baran, 2009). In turn, this lack of accurate global observational constraints leads to critical shortfalls for evaluation efforts of predictions of ice cloud properties and processes in numerical weather forecast and climate models (Lohmann et al., 2007).

The particle size distribution (PSD), which describes the number concentration of ice crystals as a function of their size, impacts most microphysical and radiative properties of clouds. This makes the PSD a central parameter in remote-sensing retrieval techniques (Yang et al., 2001; Vidot et al., 2015). PSD shapes can greatly vary, as they are influenced by formation and growth mechanisms dictated by the environment in which the cloud developed (Heymsfield and McFarquhar, 2002). For instance, the cloud origin (e.g. in-situ or liquid-origin; Krämer et al., 2016) was identified as having a major influence on PSD shapes for cirrus (Luebke et al., 2016). However, the exact drivers to PSD shapes remain poorly understood and their global variability only addressed by few modelling studies (e.g. Gasparini and Lohmann, 2016). For this reason, retrieval techniques must make critical simplifications regarding PSD shapes, which are commonly reflected by the use of fixed parameterizations that are universally applied globally and for all cloud types. Such recent parameterizations rely on advanced normalization procedures that aim to make PSDs more representative of a large sample of ice clouds (Delanoë et al., 2005; Field et al., 2007, also see Section 3). While this allows for more accurate representations of ice clouds in retrieval methods (e.g. Sourdeval et al., 2015, 2016), current parameterizations still have important issues. For instance, they largely rely on in-situ measurements where the distribution of small ice crystals (sizes less than 100 μm diameter) are at best very uncertain (Korolev et al., 2011), which makes them deficient to simulate remote-sensing observations sensitive to small ice and leads to erroneous retrievals, especially for estimations of the ice crystal number concentration ($N_{ice}$). Reaching more accurate estimations of $N_{ice}$ is particularly of great importance to understand ice cloud formation mechanisms and for improving their predictions in models, since this parameter is often used as a prognostic variable to predict the evolution of clouds (e.g. Seifert and Beheng, 2006).

An example of widely used application of normalized PSD parameterizations in satellite retrievals is found in the DARDAR algorithm (raDAR/liDAR; Delanoë and Hogan, 2008, 2010). DARDAR retrieves vertical profiles of ice cloud microphysical properties by using a combination of coincident measurements from the Cloud Profiling Radar (CPR) onboard CloudSat and from the Cloud-Aerosol Lidar with Orthogonal Polarization (CALIOP). Although the original algorithm does not include retrievals of $N_{ice}$, Sourdeval et al. (2018) explored the capabilities of its framework to retrieve this parameter (DARDAR-Nice). However, comparisons of $N_{ice}$ retrievals by DARDAR-Nice to recent in-situ measurements highlighted strong limitations linked to its used parameterizations of PSDs, which are in specific cases not suited to retrieve $N_{ice}$ (Sourdeval et al., 2018; Krämer et al., 2020). A main reason is that the currrent parameterization only represents a single ice crystals size mode, which can result in PSDs being significantly misrepresented when they happen to be strongly bimodal, i.e. consisting of two size modes. The bimodality of the PSDs can often occur in cirrus clouds and is a function of temperature and location within the cloud (Zhao et al., 2011), having warmer clouds more often bimodal PSDs (Jackson et al., 2015). Associated with the study of bimodality is the analysis of the effect of shattering ice crystals at the inlets of the in situ instruments that can artificially increase the concentration of small particles. This problem has been widely discussed and as a result, new inlets were designed and correction algorithms applied to minimize this effect (Korolev et al., 2011; Lawson, 2011; Krämer et al., 2020).

The present study investigates the benefits of considering a second mode in existing PSD parameterizations methods. It follows the framework proposed by Delanoë et al. (2005, 2014) (hereafter D05 and D14), which is applied on an extensive database of in-situ ice cloud properties from airborne campaigns (JULIA; (Krämer et al., 2016)), which is considered less sensitive to shattering effects. This study proposes a new set of parameterizations, based on single- and double-mode PSDs, that will be useful to improve remote-sensing retrieval methods. Section 2 describes the JULIA database and Section 3 details the PSD normalization method. Section 4 presents and analyzes the newly developed parameterizations by comparison to the original in-situ data. Section 5 concludes this study.

## 2 The JULIA Database

JULIA (JÜLich In situ Airborne database) is a compilation of in situ measurements of cirrus, mixed-phase and liquid clouds, water vapor and also other trace gases, collected at the Research Center Jülich starting in 2008 (Schiller et al., 2008; Luebke et al., 2013; Costa et al., 2017; Krämer et al., 2016; Krämer et al., 2018; Krämer et al., 2020) and continued until 2021. The measurements were taken onboard of different research aircraft and cover the tropics, mid-latitudes and the Arctic (Fig. 1). For our study, we focus only on the data of ice particles, which includes data from 11 field campaigns with measurements of ice crystals of diameters from 3 to 1000 μm taken every second. The analyzed data, i.e., the $N_{ice}$ and ice water content (IWC), was sampled with (or computed from) the NIXE-CAPS (New Ice eXpEriment: Cloud and Aerosol Particle spectrometer), which is a combination of CAS (Cloud and Aerosol Spectrometer) and a CIPg (Cloud Imaging Probe - grayscale), and combinations of the FCDP (Fast Cloud Droplet Probe) and 2D-S (Two-Dimensional Stereo), CDP (Cloud Droplet Probe) and 2D-C (Two-Dimensional Cloud) instruments (the respective used size ranges are listed in Table2). These instruments and their data processing procedures are described in the literature (e.g., Lawson et al., 2006; McFarquhar et al., 2007; Lawson, 2011; Krämer et al., 2016; Luebke et al., 2016; Baumgardner et al., 2017; Afchine et al., 2018; Krämer et al., 2020). Since the widths of the size bins of each pair of instruments are different, Krämer et al. (2022) synchronized all PSDs to the same grid with logarithmically equidistant size bins in order to facilitate inter-comparisons and the interpretation of the results. The data included in JULIA has undergone an extensive quality check process to warranty its validity. In the following section, a brief description of the campaigns and the instruments is given.

### 2.1 Selected airborne campaigns

The analyzed data covers a wide range of meteorological conditions and extend from the tropics to the polar region. In total there are 5 campaigns in the tropics (ACRIDICON-CHUVA, ATTREX, CONTRAST, POSIDON and STRATOCLIM), 3 in mid-latitudes (COALESC, ML-CIRRUS and START), one covering both mid and high-latitudes (CIRRUS-HL) and 2 in high latitudes (RACEPAC and VERDI). ACRIDICON consisted of flights over the Amazonian forest in September 2014 with the aim of studying the interaction between aerosols and deep convective clouds (Wendisch et al., 2016). During ATTREX (February-March 2014), measurements inside cirrus clouds were taken over the West tropical Pacific, around the tropical tropopause layer, avoiding convection (Thornberry et al., 2017). CONTRAST, based on Guam, took also place in the West

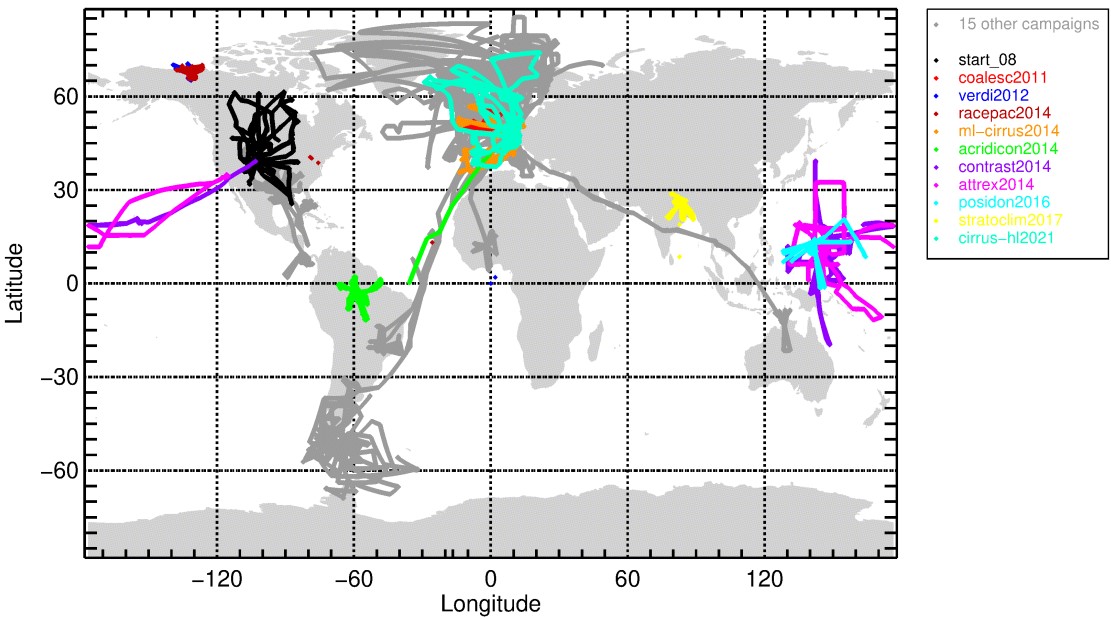

**Figure 1.** Location of all campaigns of the JULIA data base. In different colors, the campaigns used for this study, other campaigns in grey. (Figure by Nicole Spelten).

tropical Pacific (January-February 2014) to study tropical convection, oceanic processes and ozone chemistry in the upper troposphere - lowermost stratosphere (UTLS) (Pan et al., 2017). POSIDON (October 2016) was also based on Guam and among its objectives were the study of cirrus clouds and dehydration in the tropical UTLS (POSIDON: last accessed: 29 July 2022). During POSIDON three flights were near an active typhoon. The last campaign in the tropics is STRATOCLIM (July-August 2017), based in Nepal and focused on water vapor variations and upper tropospheric clouds (Krämer et al., 2020; Khaykin et al., 2022). COALESC took place during February and March 2011 and the observations were taken over the South-East coast of England and Wales. Its main focus was on stratocumulus clouds, but there are also plenty of observations within mixed-phase and cirrus clouds (Osborne et al., 2014). During March and April 2014, ML-CIRRUS took place, centered in observations over Europe and the North Atlantic focusing on processes involving cirrus (Voigt et al., 2017). START was based in Colorado, USA during April-June 2008 (Pan et al., 2010). CIRRUS-HL is a campaign from July-August 2021, based in Oberpfaffenhofen, Germany, that investigated microphysical properties and climate impact of ice clouds in high latitudes, as well as the effect of aviation (CIRRUS-HL, last accessed: 29 July 2022). VERDI (April-May 2012, Inuvik, Canada) (Costa

**Table 1.** Instruments used during the field campaigns of the JULIA data base (NIXE-CAPS as a combination of CAS: Cloud and Aerosol Spectrometer and CIPg: Cloud Imaging Probe greyscale. CDP: Cloud Droplet Probe, FCDP: Fast Cloud Droplet Probe, 2D-C: Two-Dimensional Cloud spectrometer, 2D-S: Two-Dimensional Stereo Cloud spectrometer). Range refers to the ice particle diameter. *The range of the instruments might be larger.

| Instruments | Campaigns | Aircraft | Years | Used ranges* |
|---|---|---|---|---|
| CDP + 2D-C | START, CONTRAST | HIAPER | 2008, 2014 | $3-50\,\mu m + 60-1000\,\mu m$ |
| FCDP + 2D-S | ATTREX, POSIDON | Global Hawk, WB-57 | 2014, 2016 | $3-25\,\mu m + 25-1000\,\mu m$ |
| NIXE-CAPS | COALESC, VERDI, RACEPAC | BAe-146, Polar-5/6, | 2011, 2012, 2014 | $3-25\,\mu m + 25-930\,\mu m$ |
| NIXE-CAPS | ML-CIRRUS, ACRIDICON-CHUVA | HALO | 2014 | $3-25\,\mu m + 25-930\,\mu m$ |
| NIXE-CAPS | STRATOCLIM, CIRRUS-HL | Geophysica, HALO | 2017, 2021 | $3-25\,\mu m + 25-930\,\mu m$ |

et al., 2017) aimed to investigate, among other goals, the radiation budget of ice clouds in the Arctic and the influence of convective transport in the upper troposphere. During this campaign a persistent anticyclone was present and favored the formation of persistent stratus (Klingebiel et al., 2015). RACEPAC (Costa et al., 2017) is the follow up campaign of VERDI

and took place in April-May 2014.

In Table 1 a summary of the used instruments for each campaign is given. The NIXE-CAPS is formed by CAS and CIPg that combined cover particles with diameters from 0.61 to 930 μm. To avoid overlap of particle sizes, the PSDs from NIXE-CAPS are merged between 20 and 25 μm. Detailed explanations about the data analysis methods, including position of the instrument in the aircraft and shattering effects, are given by Meyer (2013); Krämer et al. (2016); Luebke et al. (2016); Costa (2017). Other

instruments are the light scattering sensors CDP that measures concentration for particles with diameters between 2 and 50 μm (McFarquhar et al., 2007) and the FCDP for particles between 1 and 50 μm (Baumgardner et al., 2017). The optical imaging cloud probe 2D-C spectrometer (Baumgardner et al., 2017) and the 2D-S spectrometer are used to reconstruct cloud particle shapes and sizes between $25-800\,\mu m$ and $5-1280\,\mu m$, respectively. The 2D-S includes tips and software to reduce shattering effects (Lawson et al., 2006; Lawson, 2011). FCDP and 2D-S PSDs are merged between 20 and 25 μm and the ones from

CDP and 2D-C at 55 μm. For more information about the campaigns and the instruments the reader is also referred to Costa (2017); Krämer et al. (2020).As mentioned in Sect. 1, shattering of the ice particles during the measurements would increase the number of small particles and cause an artificial bimodality in the PSDs. However, as presented in the above references, major efforts were made in the development of antishatter probe tips and particle interarrival time algorithms that have resulted in a successful minimization of the shattering of ice particles (see e.g. Krämer et al., 2020). Therefore, we are confident that

the bimodality present in the JULIA database is not due to distorted microphysical properties of the PSDs.

## 2.2 Ice crystal number concentration and ice water content computation

$N_{ice}$ is the sum of the ice crystal number concentration of each size bin. The IWC is obtained following:

$$IWC = \sum_{i=1}^{n} m_i \cdot \Delta N_{\text{ice, i}}, \tag{1}$$

where $m_i$ and $\Delta N_{ice, i}$ are the mass and number concentration of the ice crystals in the $i$th size bin, respectively. The mass $m$ is computed using the mass - dimension ($m$ - $D$) relation of the form $m = a \cdot D^b$ described in Krämer et al. (2016), which is based on the modified $m$-$D$ relation of Mitchell et al. (2010). The coefficients $a$ and $b$ depend on the ice crystal size. The used $m$ - $D$ relation was compared in Afchine et al. (2018) with other $m$ - $D$ relations from the literature (covering, depending on the $m$ - $D$ relation, temperatures between $-65°$ C and $0°$ C) and also with the measurements from total water instruments showing good agreement for cirrus clouds. For temperatures higher than the cirrus range, we are aware that the uncertainties of the derived IWC are larger than at lower temperatures. However, we use the same $m$ - $D$, since as shown in Afchine et al. (2018), the differences between the compared $m$ - $D$ relations is small, even when considering those derived for higher temperatures.

## 3 Normalization method

The method used to fit the PSDs is based on the methodology described by D05 and D14, which is an adaptation of the normalization method for raindrop spectra introduced by Testud et al. (2001). It consists of computing several moments of the measured in situ PSDs, using them to normalize the in situ PSDs and then fitting the normalized PSDs to a certain function. This universal function can then be used to obtain microphysical and optical properties of cirrus clouds. Here, we present the key points of the method, for a complete description, the reader is referred to the aforementioned references.

The first step is to compute the equivalent melted diameter ($D_{\text{eq}}$) following:

$$D_{\text{eq}} = \left( \frac{6 \cdot m(D)}{\pi \rho_{\text{w}}} \right)^{1/3}, \tag{2}$$

where $m(D)$ is obtained with the $m$ - $D$ relation indicated in Sect. 2.2 and $D_{eq}$ is in units of meters.

Using the measured PSD, the moments of the distribution are computed as follows:

$$M_{\text{n}} = \int_{0}^{\infty} D^n N_{ice}(D) dD \approx \sum_{D=min}^{D=max} D^n N_{ice}(D) \Delta D, \tag{3}$$

with $n$ the moment order, $N_{ice}(D)$ the ice number concentration for the size bin $D$ and $\Delta D$ the width of the corresponding size bin.

D05 and D14 use the volume-weighted diameter $D_m$ to scale the PSD in the size space and the intercept parameter $N_0^*$ in the number concentration space. These parameters can be defined in terms of the third and fourth moment of the PSD by:

$$D_m = \frac{\int_0^\infty N_{ice}(D_{eq}) D_{eq}^4 dD_{eq}}{\int_0^\infty N_{ice}(D_{eq}) D_{eq}^3 dD_{eq}}, \quad \text{and} \tag{4}$$

$$N_{ice}(D_{eq}) = N_0^* F(D_{eq}/D_m),$$ (5)

considering that $N_{ice}(D_{eq})dD_{eq} = N_{ice}(D)dD$ for a given size bin. $F$ is a modified gamma function that describes the shape of the normalized PSDs and, as first given by D05 is:

$$F(\alpha,\beta,X) = \beta \frac{\Gamma(4)}{4^4} \frac{\Gamma(\frac{\alpha+5}{\beta})^{4+\alpha}}{\Gamma(\frac{\alpha+4}{\beta})^{5+\alpha}} X^\alpha exp\left[ -\left( X \frac{\Gamma(\frac{\alpha+5}{\beta})}{\Gamma(\frac{\alpha+4}{\beta})} \right)^\beta \right]$$ (6)

$F$ is defined by four parameters. $N_0^*$ and $D_m$ (through $X = D_{eq}/D_m$) that change for each PSD and $\alpha$, $\beta$ that are fixed and can be found by computing a cost function. The normalization is applied to each individual PSD for all campaigns.

The $\alpha$ and $\beta$ that best fit the normalized measured PSD are chosen using an in situ database (in our study the JULIA dataset) and a least square regression linear fit on moments of the PSD (Field et al., 2005, 2007; Delanoë et al., 2014). Following D14, we use a combination of a low and a high moment of the PSD. However, unlike previous studies, we here aim at improving the direct prediction of physical parameters of the PSDs and therefore minimize $N_{ice}$ and IWC via the cost function. This cost function, $J$, is commonly used to quantify the consistency between predicted and in-situ parameters. Considering tendency of both IWC and $N_{ice}$ to follow a log-normal distribution, we used the logarithmic values of these two parameters when computing $J$:

$$J = \sum_{i=1}^{n} (J_{N_{ice}} + J_{IWC})$$ (7)

with $J_{N_{ice}}$ and $J_{IWC}$ as:

$$J_{N_{ice}} = \left( 1 - \frac{\log(N_{param}(\alpha,\beta))}{\log(N_{insitu})} \right)^2$$ (8)

$$J_{IWC} = \left( 1 - \frac{\log(IWC_{param}(\alpha,\beta))}{\log(IWC_{insitu})} \right)^2$$ (9)

where $n$ is the total number of PSDs, $N_{insitu}$ is the sum of the ice crystal number concentration from the in-situ database in each size bin and $IWC_{insitu}$ is derived using Eq. 1. $N_{param}$ is computed by integration of the size distribution $N_{ice}(\alpha,\beta,D)$ obtained from the normalized function F and using only the size bins that are present in the in situ data. $IWC_{param}$ is derived by applying Eq. 1. The objective of the cost function approach is to find the optimal coefficient pair of $\alpha$ and $\beta$ that will minimize $J$.

## 4 Results

### 4.1 Original and normalized PSDs

As cirrus we understand all clouds colder than 235 K (Krämer et al., 2016). In the temperature range just below 235 K, the clouds may originate as mixed-phase clouds ascending from lower altitudes, undergoing complete glaciation at $\geq$ 235 K. This physical definition of cirrus is based on the ice formation mechanism, which is on the one hand the just mentioned complete glaciation of liquid clouds (liquid origin cirrus) and on the other hand cirrus that form directly as ice (in-situ origin cirrus). The analyzed in-situ data is limited to temperatures lower than 255 K. This choice was made to minimize the selection of water droplets in mixed-phase clouds, since below this temperature most of the water droplets are glaciated. No differentiation between contrail and natural cirrus was made in our analysis. In total, 542 719 PSDs were analyzed ($\approx$ 151 h). Of the total number of analyzed PSDs, about 9.8 % are found at temperatures between 235 K and 255 K (i.e. mixed-phase regime). The individual contribution of each campaign is registered in Table 2.

**Table 2.** Contribution in % of each campaign to the total analyzed data considering only T < 255 K. The columns indicate the name of the campaign, the amount of hours analyzed and the number of PSDs of each campaign and their contribution in %. Total number of PSDs: 542 719. Total number of cumulative hours: 150.76.

| Campaign | Hours | Num. PSDs | % |
|---|---|---|---|
| ACRIDICON | 11.62 | 41833 | 7.71 |
| ATTREX | 30.45 | 109605 | 20.19 |
| CIRRUS-HL | 29.38 | 105410 | 19.42 |
| COALESC | 11.2 | 40325 | 7.43 |
| CONTRAST | 25.34 | 91220 | 16.81 |
| ML-CIRRUS | 20.54 | 73934 | 13.62 |
| POSIDON | 11.72 | 42178 | 7.77 |
| RACEPAC | 0 | 15 | 0.002 |
| START | 4.02 | 14473 | 2.67 |
| STRATOCLIM | 6.03 | 21697 | 4 |
| VERDI | 0.56 | 2029 | 0.37 |

Figure 2 (right column) shows how PSDs contain information about the characteristics of clouds and their history and may differ according to dynamical conditions. These differences can be seen in the frequency distributions of the concentrations of the PSDs of each campaign. For example, very high number concentrations ($> 1$ cm$^{-3}$) of the smallest ice crystals can be due to the presence of contrails (e.g. during COALESC, ML-CIRRUS or CIRRUS-HL). Another example, during several flights of ACRIDICON-CHUVA and CIRRUS-HL strong convection was present (i.e. fast updrafts). During fast updrafts, air masses can be lifted and reach temperatures lower than 235 K and if the conditions are favourable, ice nucleation will take place and result

in the formation of small ice crystals. Also during strong updrafts small supercooled liquid water droplets glaciate, giving as a result a large number of small ice crystals. In the PSDs that is translated in an increase in number concentration between $1\times10^{-2}$ - $1\,\mathrm{cm}^{-3}$ for ice crystal sizes smaller than $\approx$ 20 µm. Another indicator of cirrus clouds that have their origin at T < 235 K is the presence of large ice crystals that come from heterogeneous drop freezing ($\approx$ 200 µm, more noticeable for particles > 500 µm that reach a number concentration of $\approx$ $1\times10^{-4}\,\mathrm{cm}^{-3}$). A more detailed analysis of the characteristics of the PSDs according to dynamical conditions will be given in Krämer et al. (2023).

Figure 2 (left column) shows the results of applying the normalization method described in Sect. 3 for a selection of campaigns (see figure legend for detail). As described in D05 and D14, the normalization approach removes some natural variability of PSDs and therefore narrows the data down to smaller size and concentration areas (mostly around $D_{eq}/D_m = 1$), in comparison to the original observed PSD. Another visible feature of the normalization is that the normalized spectra look similar for all campaigns, although there are some differences depending on the measurement location and the cloud processes involved (Delanoë et al., 2005; Field et al., 2007). Overall, these features are also visible in our study, including as well the larger variability of the PSDs tail, which is linked to the temperature. In D14 it is explained that for very low temperatures (less than -60°C) the tail vanishes and for higher temperatures the variability of $D_m$ is larger, since the range of possible ice crystal sizes increases due to ice particles coming from different microphysical processes.

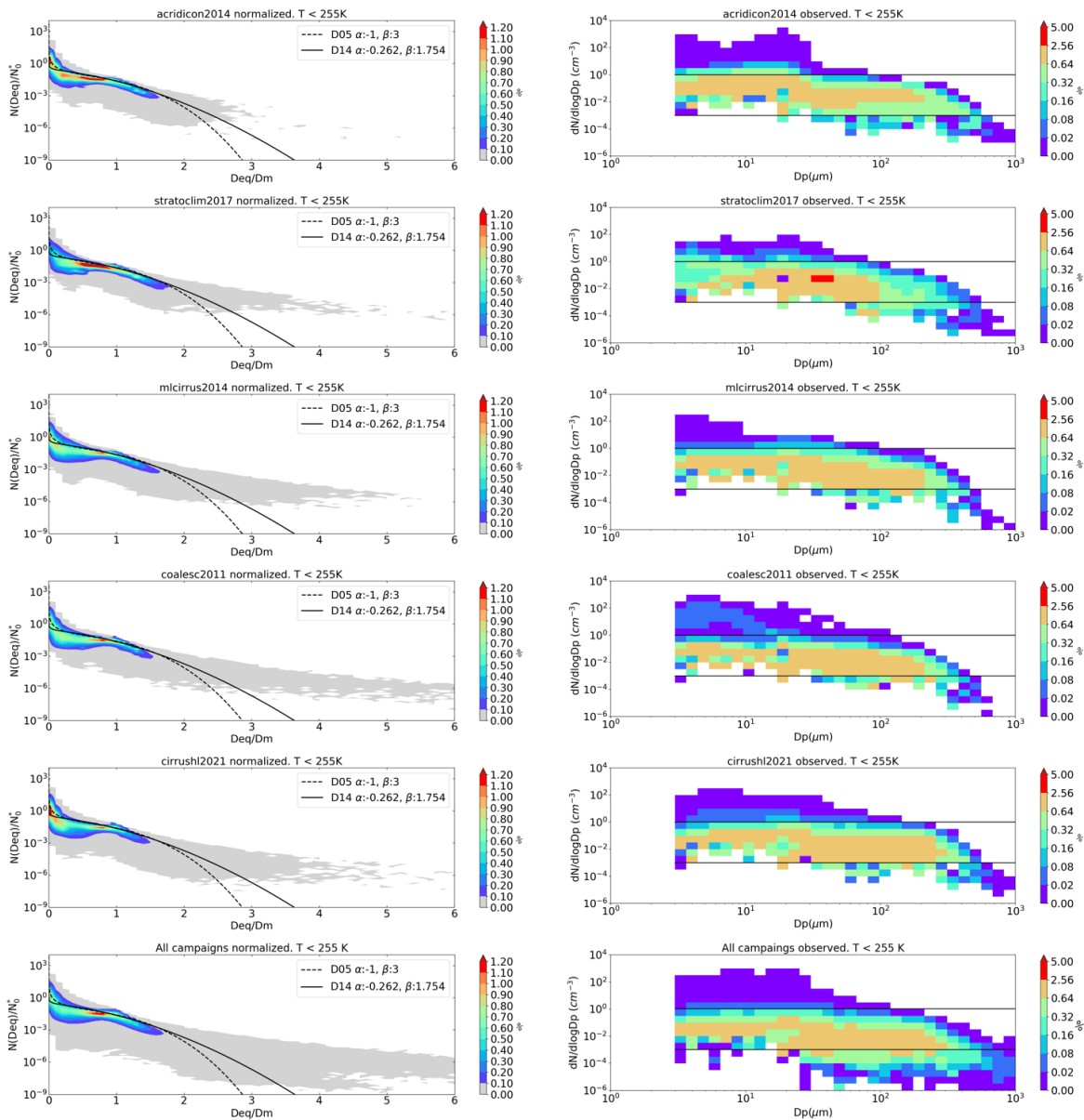

**Figure 2.** The right column shows size-resolved occurrence frequencies of ice crystals of the observed PSDs and the left column their corresponding normalized PSDs for a selection of campaigns covering the tropics (ACRIDICON-CHUVA, STRATOCLIM), mid-latitudes (COALESC, ML-CIRRUS, CIRRUS-HL) and polar region (CIRRUS-HL); the bottom panels show the combination of all campaigns. In the panels of the left column, the black dashed and solid lines corresponds to the proposed parameterization by D05 and D14, respectively.

## 4.2 Investigation of optimal PSD parameters

As described in Sect. 3, a cost function minimization approach is used in this study to find an optimal pair of $\alpha$ and $\beta$ coefficients needed to define the normalized PSD parameterizations that will fit in-situ measurements from JULIA. We seek to optimize the representation of physical properties, IWC and $N_{ice}$ (see Eq. 7 to Eq. 9), as opposed to optical properties as for instance in D05 and D14. This is done with the intent to produce a PSD parameterization that can be universally used for retrievals of $N_{ice}$ and IWC using a wide variety of instruments.

In D05 and D14, whose parameterizations are here used for comparison purposes, the lidar extinction coefficient and radar reflectivity factor are used to find the best parameters for the modified gamma function. The selected $\alpha$ and $\beta$ pairs in the study by D14 yield very similar normalized PSD shapes for each campaign, with almost a complete overlap of the normalized function for $D_m/D_{eq}$ between 0.5 and 1.5 (see Fig. 9 of D14). Following Fig. 9 of D14, we also plotted the normalized function for each campaign (i.e $N_{ice}/N_0^*$ vs $D_m/D_{eq}$) (Fig. 3). Figure. 3a) shows the normalized function for each campaign when selecting the optimal parameters using in the cost function both the log(IWC) and the log($N_{ice}$). Figure 3b shows the result of only using log(IWC) for the cost function and Fig. 3c the result of only using log($N_{ice}$). Figure 3d is an overview of the selected optimal parameters $\alpha$ and $\beta$ for each campaign (in different colors) and for all together (black circle, JULIA 1M in Table 3). In our study, although the normalized functions also cluster around $D_m/D_{eq}$ values between 0.5 and 1.5, the overlap is not as pronounced as in D14 and the spread in the $\alpha$ and $\beta$ values is also larger (Fig. 3d). To try to obtain a more compact cluster of selected coefficient pairs, we divided the PSDs into ice crystals smaller and larger than $50\,\mu m$ (not shown) (JULIA small and JULIA large in Table 3). Splitting the PSDs modifies the spread, but our cluster in any case looks as compact as in D14. This might be due to the selected parameters to compute $J$. Whereas D14 uses optical parameters, i.e., visible extinction and reflectivity, we use physical parameters, i.e., $N_{ice}$ (which is very sensitive to temperature) and IWC. Moreover, although we follow the indication of applying one moment sensitive to the small particles and another one to the large particles, the selected moments are not the same. In D14 the parameters are proportional to the second and approximately sixth moment of the distribution and in our study to the zeroth moment and between the second and third moment. To summarize, the parameterizations differ in the data used to compute each of them, the m-D relationship used and how the parameters of the modified gamma function were obtained. Table 3 gives an overview of the selected coefficients for each parameterization and in Sect. 4.3 we discuss the characteristics of each of them.

**Table 3.** Best $\alpha$ and $\beta$ coefficients for each parameterization. D05 is from Delanoë et al. (2005) and D14 from Delanoë et al. (2014). D14 was obtained by Delanoë et al. (2014) using extinction and reflectivity in the cost function. All other parameterizations are based on the JULIA database and the use of Eq.7. JULIA 1M refers to one mode, JULIA large to D $\geq$ 50 µm, JULIA small to D < 50 µm and JULIA 2M to the combination of the small and large modes.

| Parameterization | $\alpha$ | $\beta$ |
|:---:|:---:|:---:|
| D05 | $-1$ | 3 |
| D14 | $-0.262$ | 1.754 |
| JULIA 1M | $-0.945$ | 0.886 |
| JULIA large | 0.968 | 3.307 |
| JULIA small | $-0.968$ | 5.225 |
| JULIA 2M | JULIA small + JULIA large | |

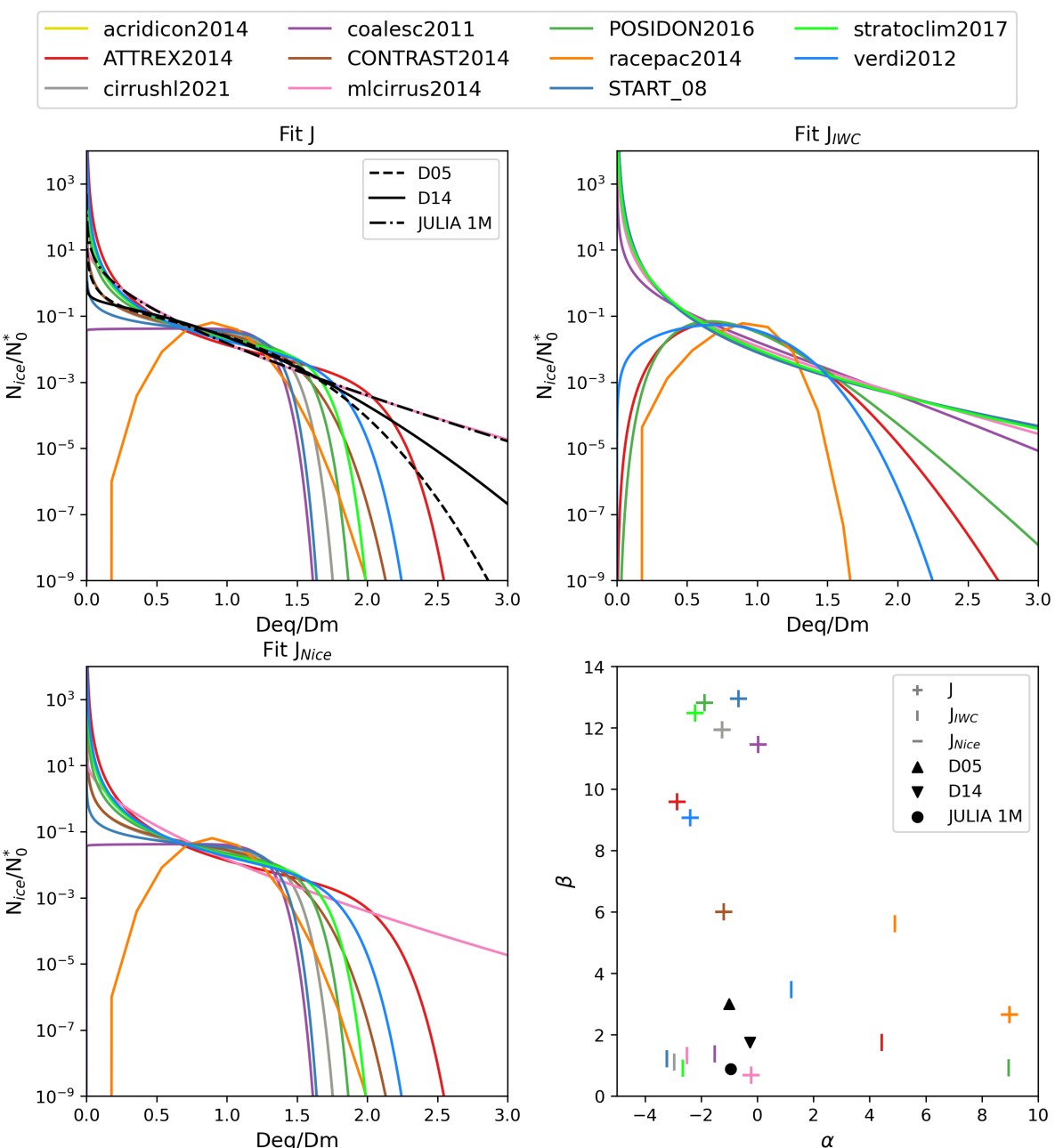

**Figure 3.** (a-c) normalized functions for each campaign after selecting the best pair of coefficients. (a) is the result of computing the cost function $J$ using IWC and $N_{ice}$, (b) of using only IWC to compute $J$ and (c) of using only $N_{ice}$. (d) shows the selection of best coefficients pair for each campaign and the coefficients for the parameterization by D05 (black upwards triangle), the parameterization proposed in D14 (black downwards triangle) and the new one (black circle). For this case, the horizontal line symbol in (d) is coincidental with the symbol of the crosses. JULIA 1M refers to the monomodal parameterization obtained in this study. A detailed overview of the coefficients of each parameterization is given in Table 3. Only PSDs measured at T < 255 K are considered.

## 4.3 One mode vs. two modes parameterization

Ice crystals smaller than about 50 μm present a greater variability in the PSDs as the larger ice crystals (see Fig. 2, right column), because they correspond to the regime of newly formed ice crystals that quickly grow to larger, sedimenting sizes. In cirrus clouds, riming and secondary ice production do not play any role and aggregation, at the lowest temperatures, is nearly negligible. These processes are of importance for mixed-phase clouds, which, as mentioned in Sect. 4.1 entail 9.8 % of the analyzed data. In Jackson et al. (2015) it was discussed that at temperatures lower than $-45\,°C$ the growth of the ice crystals is likely due to depositional growth whereas sedimentation and aggregation are less significant. For higher temperatures, the ice particles also grow by vapor deposition, but sedimentation from above can also be a source of large particles that causes bimodality (Zhao et al., 2011). Another process that can lead to bimodality is a two-step ice nucleation. Initially, the few heterogeneously nucleated ice crystals may grow to larger sizes, followed by homogeneous nucleation of more and smaller ice crystals. However, the main reason for the bimodality of cirrus PSDs is the superposition of in-situ origin and liquid origin cirrus. Ice crystals of liquid origin are significantly larger than those of in-situ origin because they stem from lower altitudes where there is more water to allow them to grow to large sizes, especially since only very few drops out of a liquid cloud freeze so the available water vapor is deposited only among them. Examples of measured bimodal PSDs are shown in Appendix A. In our study we compare the results of using one or two size modes, i.e., a parameterization for small particles (D < 50 μm) combined with another parameterization for large particles (D ≥ 50 μm). This cutting diameter agrees well with the division between the small and large modes when plotting the median PSD of all data (not shown).

Figure 4 shows a comparison between the median of the percentage error between the single PSDs for different parameterizations. In the upper row, Fig. 4a shows the comparison between the parameterizations JULIA 1M and JULIA 2M proposed in this study. Figure 4b shows the parameterizations from the literature and the JULIA 2M. The bottom row (Fig.4c-d) is the same as the upper row but adding in shaded color the corresponding region between the percentile 25 and percentile 75 of each parameterization to illustrate their variability. It is important to notice when considering the large values of the percentage error, that between the number concentration of the largest ice crystals and the number concentration of the smallest crystals, there can be a difference of six orders of magnitude (Fig. 2).

From Fig. 4a and 4c it is clear, that the bimodal parameterization (JULIA 2M: black) is closer to the observations than the proposed mono-modal (JULIA 1M: yellow) and that the variability is reduced for the bimodal parameterization. Between the bimodal parameterization and the ones from the literature (Fig. 4b and 4d), the clearer improvement is for ice crystals between 20 and $\approx 110\,μm$. For the small particles (smaller than 20 μm), D14 (dark blue) presents the highest deviation. This parameterization underestimates the number concentration of particles in this size range. In the case of D05 (red), the number concentration is also underestimated, but in less extent. The bimodal JULIA 2M (black) is the only parameterization that slighlty overestimates the number concentration of the smaller particles. Figure 4d shows that the variability is considerable for all parameterizations and the peak at around 20 μm is most probably caused by the merging of two instruments at this size. A detailed analysis by temperature of the comparison from Fig. 4b, presented in Fig. 5, shows that the lowest temperatures ($-90°\,C$ and $-60°\,C$) have the lowest deviation between observations and parameterizations for the smaller particles. For

larger particles, all parameterizations tend to overestimate the concentration (PSD percentage error $< 0$), with JULIA 2M under-estimating the concentration for particles larger than $\approx 600\,\mu m$. For temperatures between $-60$ and $-50^\circ$ C, where bimodality starts playing an important role, the parameterization from D05 (red) and D14 (blue) overestimate the number concentration for particles between 20 and $110\,\mu m$. This overestimation is also observed between $-50^\circ$ C and $-20^\circ$ C. However, the tendency for particles smaller than $20\,\mu m$ for D05 and D14 is to underestimate their concentration. For this range of temperatures, the bimodal parameterization is closer to the measured number concentrations, especially between 20 and $110\,\mu m$. As indicated by the median of the percentage error for particles smaller and larger than $50\,\mu m$, using a bimodal parameterization improves the representation of both the small and large modes, improving the large mode especially for higher temperatures.

In terms of IWC (not shown), correlation plots show that there are no significant differences between the results of the parameterizations (all parameterizations have a correlation factor of 1.0 and a RMSE of 0.18 or 0.19 when considering $T < 255\,K$). There is a slight underestimation (about 2 %) of the IWC for values between about $1 \times 10^{-7}\,\mathrm{gm}^{-3}$ and $1 \times 10^{-5}\,\mathrm{gm}^{-3}$ and an overestimation between about $1 \times 10^{-3}\,\mathrm{gm}^{-3}$ and $1\,\mathrm{gm}^{-3}$ (about 7 %). Since all parameterizations have a similar behaviour for the large particles and IWC is sensitive to large particles ($\gtrsim 300\,\mu m$), this result was expected.

In terms of $N_{ice}$, correlation plots between the measurements and parameterizations (Fig. 6) show that the JULIA 2M parameterization (lower row) significantly reduces the spread seen in the monomodal D05 parametrization (upper row) and that the highest frequency for D05 is found slightly above the one-to-one line. Considering temperatures lower than 255 K, the root mean square error (RMSE) is 0.34 for D05 and 0.21 for the JULIA 2M parameterization (Fig. 6c and Fig. 6f, respectively). The bimodal parameterization presents a slightly higher correlation factor (0.98 vs. 0.96) than D05. Binning this data into $10^\circ$ C temperature intervals between $-90^\circ$ C and $-60^\circ$ C (Fig. 6a, b, d, e) , where there is only one mode, JULIA 2M shows the lowest RMSE, with 0.22 vs 0.30 of the monomodal D05. For temperatures between $-60^\circ$ C and $-20^\circ$ C, where two modes are clearly visible, D05 has a RMSE of 0.36 vs. a RMSE of 0.20 of the two-modes parameterization. A comparison in temperature intervals of $10^\circ$ C (not shown), gives the same conclusions as already discussed, similar results for the lowest temperatures, being the two modes parameterization slightly closer to the observations, and better results of the two-modes parameterization for the higher temperatures.

These analysis confirm what was hinted by Sourdeval et al. (2018), i.e., defining a two-modes parameterization instead of one-mode improves the reconstruction of PSDs and retrieval of $N_{ice}$ for higher temperatures because it adjusts better to the bimodal shape of the PSDs, when it occurs.

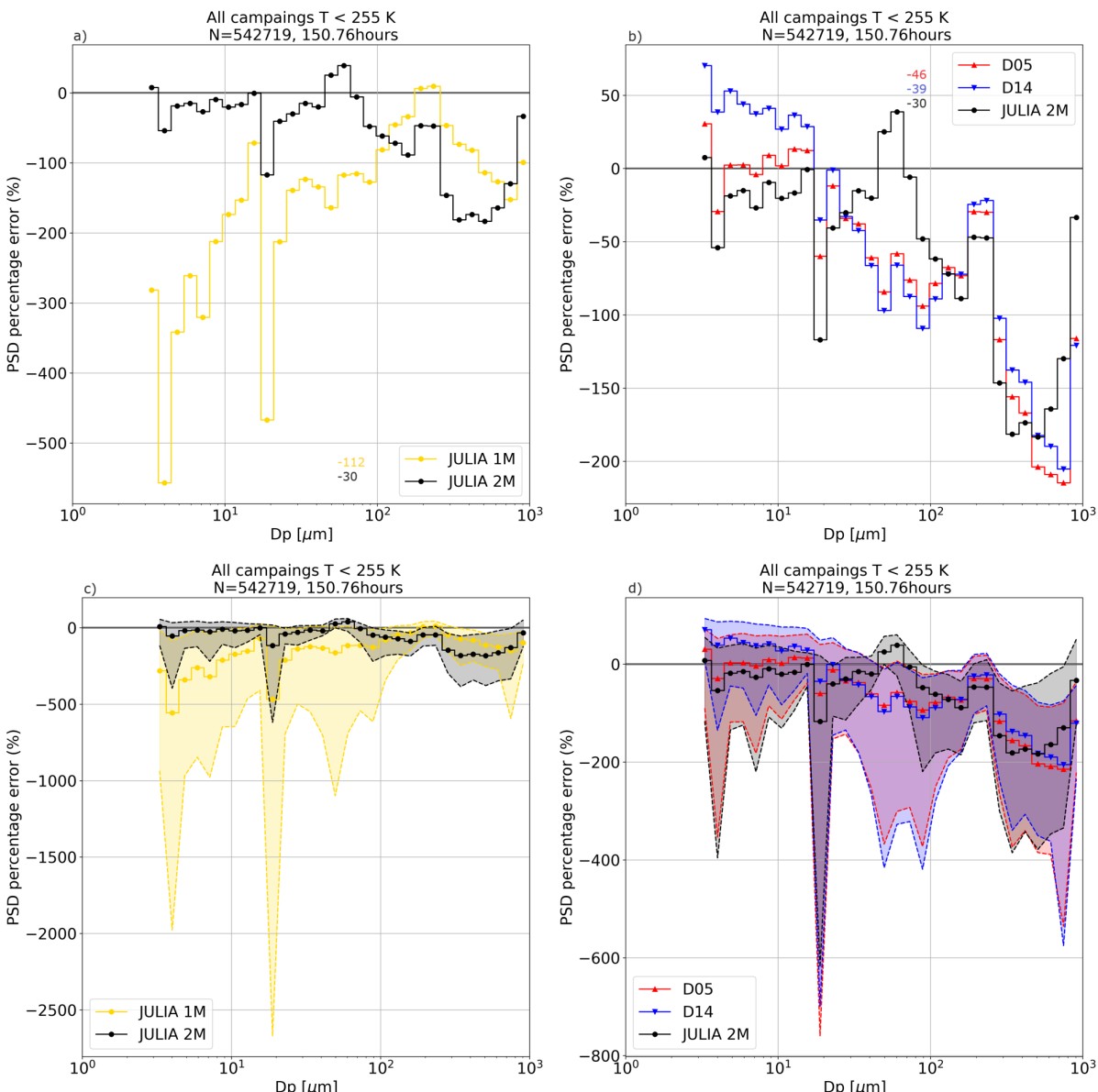

**Figure 4.** Median of the percentage error between the single observed PSDs and their corresponding parameterized PSDs. (a, c) Comparison between one mode (JULIA 1M: yellow) and two mode parameterization (JULIA 2M: black); (b, d) comparison between parameterizations from the literature (D05: red, D14: blue) and the two-modes parameterization (JULIA 2M: black) of this study. The shadow region in the lower panels correspond to the region between the percentile 25 and percentile 75. x-axis represent the size bins in μm. A detailed overview of the coefficients of each parameterization is given in Table 3. The numbers inside the panels correspond to the median percentage error over the complete size range.

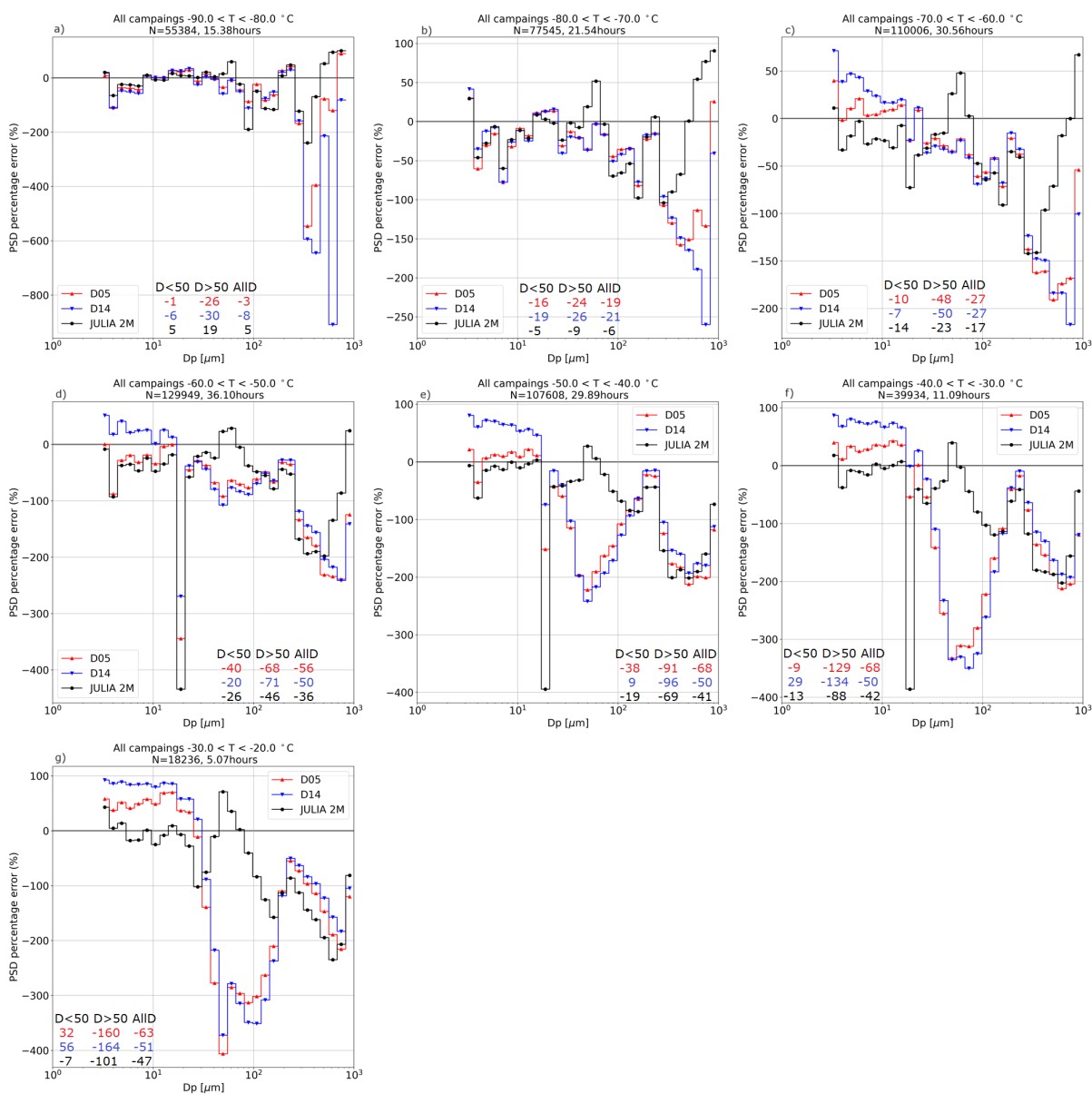

**Figure 5.** Median of the percentage error between the single observed PSDs of all campaigns and their corresponding parametrized PSDs, as in Fig. 4c, but for $10°$ C temperature intervals. The parameterizations from the literature are: D05 in red and D14 in blue. JULIA 2M (in black) is the proposed new bimodal parameterization. x-axis is the size bins in μm. A detailed overview of the coefficients of each parameterization is given in Table 3. The numbers inside the panels correspond to the median percentage error over the specified size range.

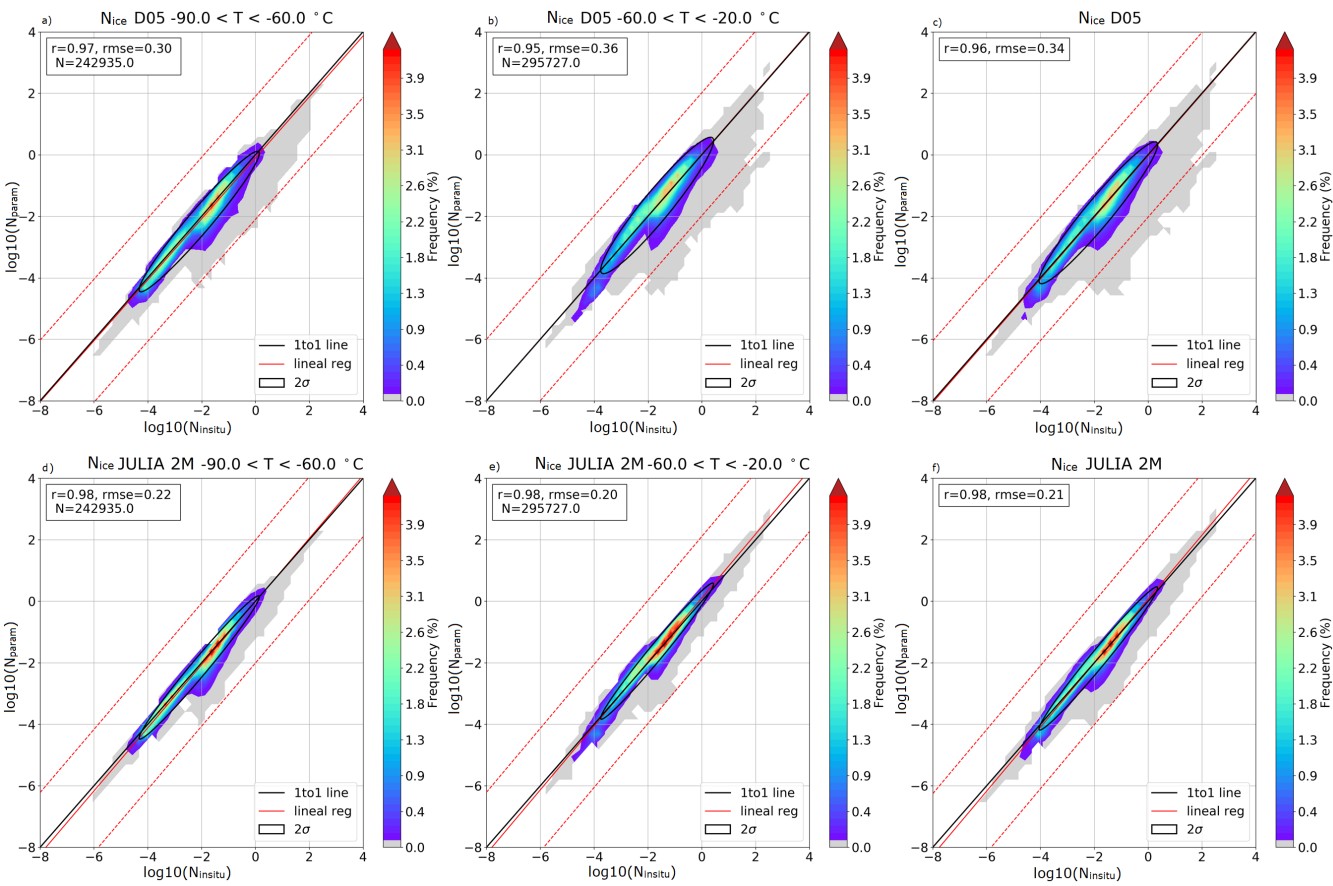

**Figure 6.** Comparison between the in-situ $N_{ice}$ and parameterized $N_{ice}$ for temperatures between $-90°$ C and $-60°$ C (a) D05, (d) JULIA 2M), temperatures between $-60°$ C and $-20°$ C (b) D05, (e) JULIA 2M) and temperatures lower than 255 K (c) D05, (f) JULIA 2M). The y-axis is the logarithm of the parameterized variable and the x-axis the logarithm of the observed one. Color code indicates frequency (%). The black line corresponds to the 1 to 1 line, the solid red line to the regression line and the dashed red lines to the regression line shifted a factor of $\pm 2$. The black ellipse is the $2\sigma$ area. A detailed overview of the coefficients of each parameterization is given in Table 3.

## 5  Conclusions

In the present study, recent airborne in-situ measurements of ice PSDs from the JULIA database were used to assess the ability of existing PSD parameterizations to represent $N_{ice}$ and IWC, and to investigate the added-value of considering two-mode PSDs. One of the main advantages of JULIA with respect to other datasets used for constructing current PSD parameterizations is that it includes observations of ice crystals down to $3\,\mu m$, which are consistently processed for all field campaigns in particular to minimize the impact of ice shattering effects. To find a new parameterization, we have followed the method by Delanoë et al. (2014), which consists of normalizing the PSDs and fitting them to a modified gamma function whose coefficients are chosen by minimizing a cost function $J$. The variables chosen to define the cost function were IWC and $N_{ice}$. We found that considering a possible bimodality of PSDs by combining two parameterizations, one for particles with $D < 50\,\mu m$ and one for $D \geq 50\,\mu m$, yields better results when compared to the observations than the hitherto used monomodal parameterizations. Considering a second mode improves the PSD prediction of both small and large ice crystals despite the large measurement uncertainties associated with the latter. The variability of the retrieved around the observed PSDs is reduced across all analyzed temperatures and there is a better fit, especially for ice particles between 20 and $\approx 110\,\mu m$ and for temperatures between $-60$ and $-20°C$. For this temperature range, the RMSE for the retrieved $N_{ice}$ is reduced from 0.36 to 0.20. In conclusion, we propose here a new bimodal ice particle PSD parameterization including ice crystals smaller than $50\,\mu m$. An important next step would be to test the feasibility of implementing two parameterizations, one for smaller particles and another for larger particles, in the retrieval algorithms of remote sensing instruments.

## Appendix A: Examples of in-situ observed bimodal ice particles PSDs

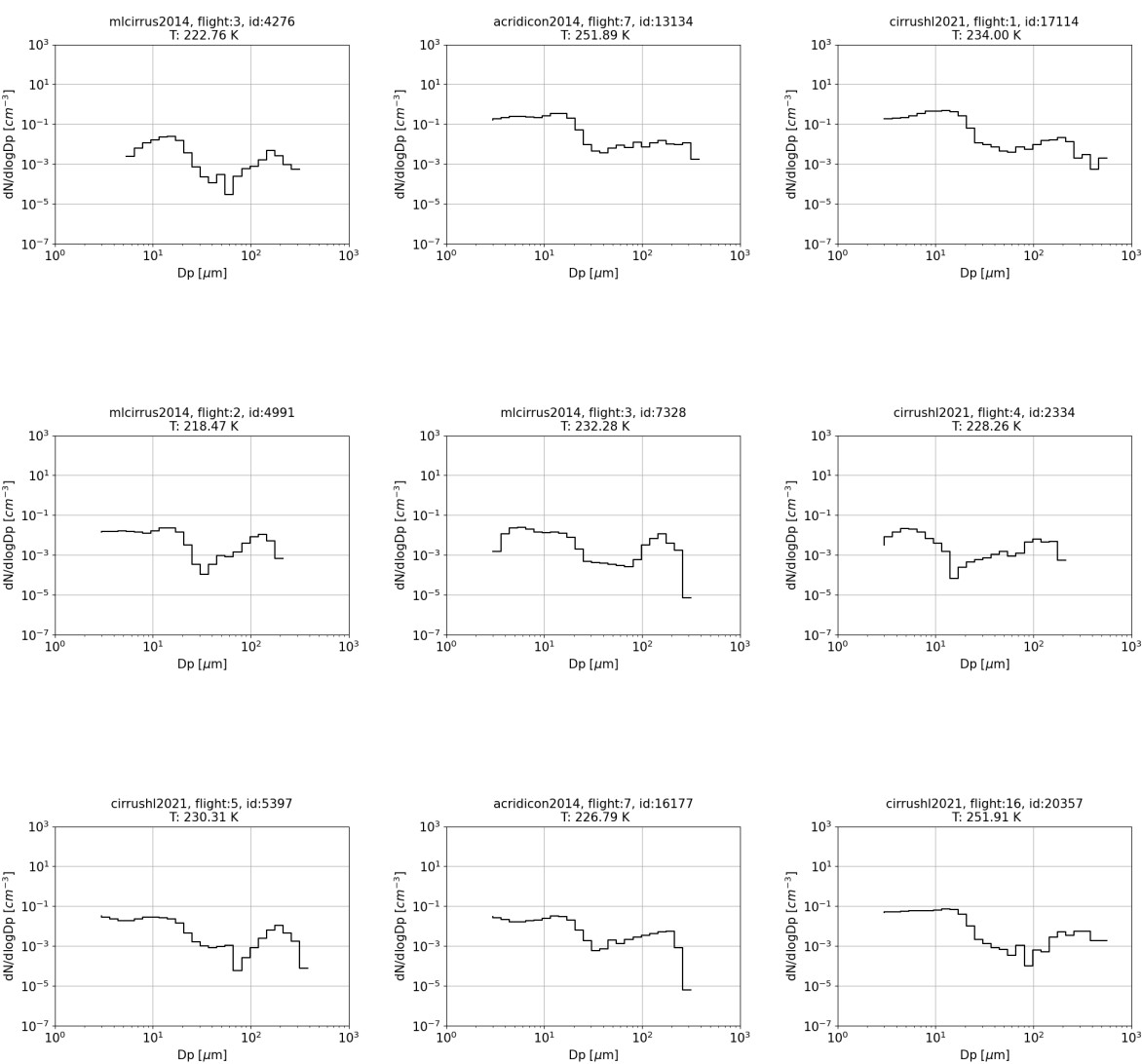

**Figure A1.** Selection of in-situ observed bimodal PSDs belonging to different airborne campaigns included in the JULIA database. In the figures the corresponding campaign, flight, number of PSD within the flight and temperature are specified. Y-axis is the concentration in $\text{cm}^{-3}$ and the x-axis is the diameter in µm.

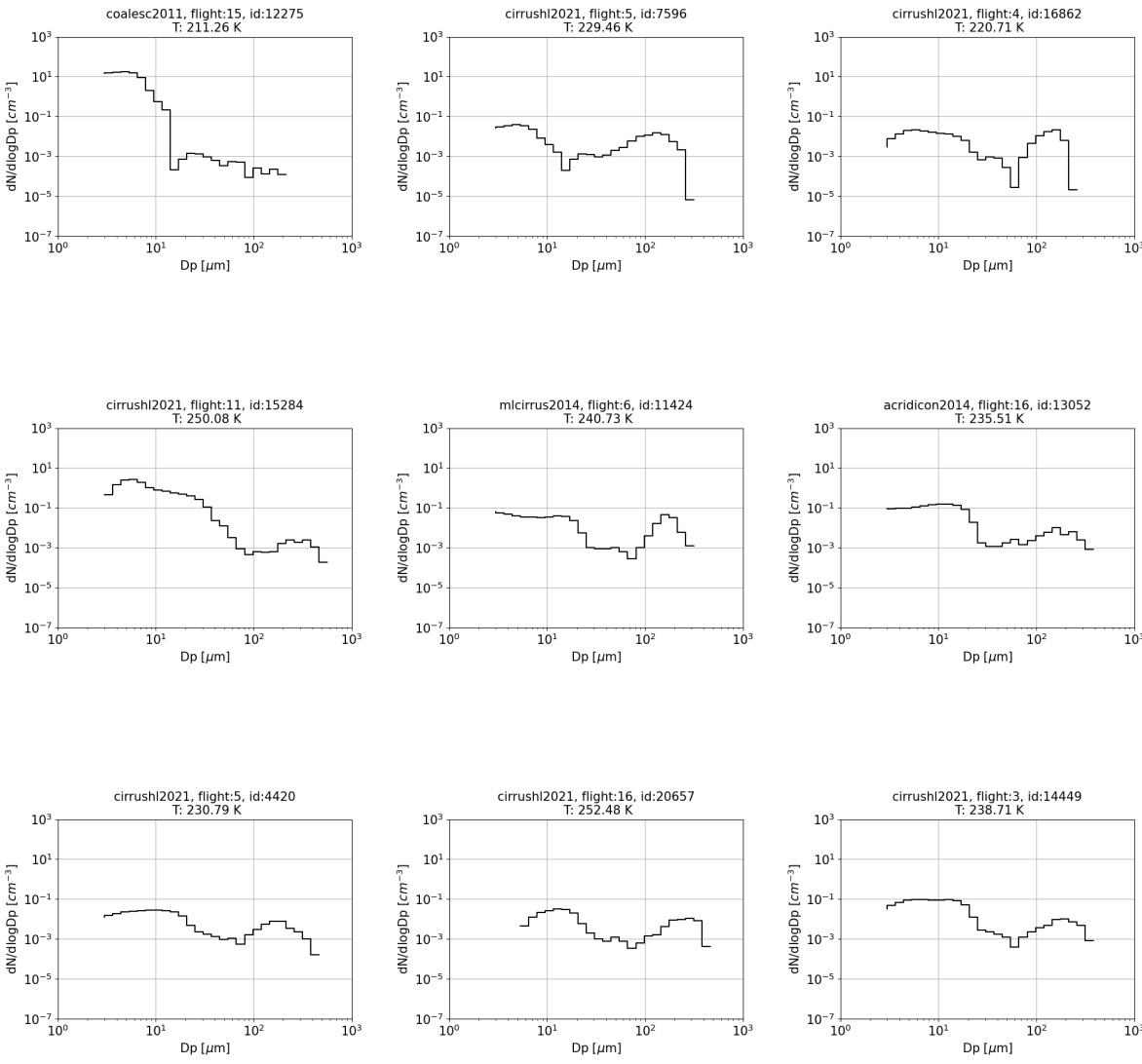

**Figure A2.** Selection of in-situ observed bimodal PSDs belonging to different airborne campaigns included in the JULIA database. In the figures the corresponding campaign, flight, number of PSD within the flight and temperature are specified. Y-axis is the concentration in $cm^{-3}$ and the x-axis is the diameter in μm.

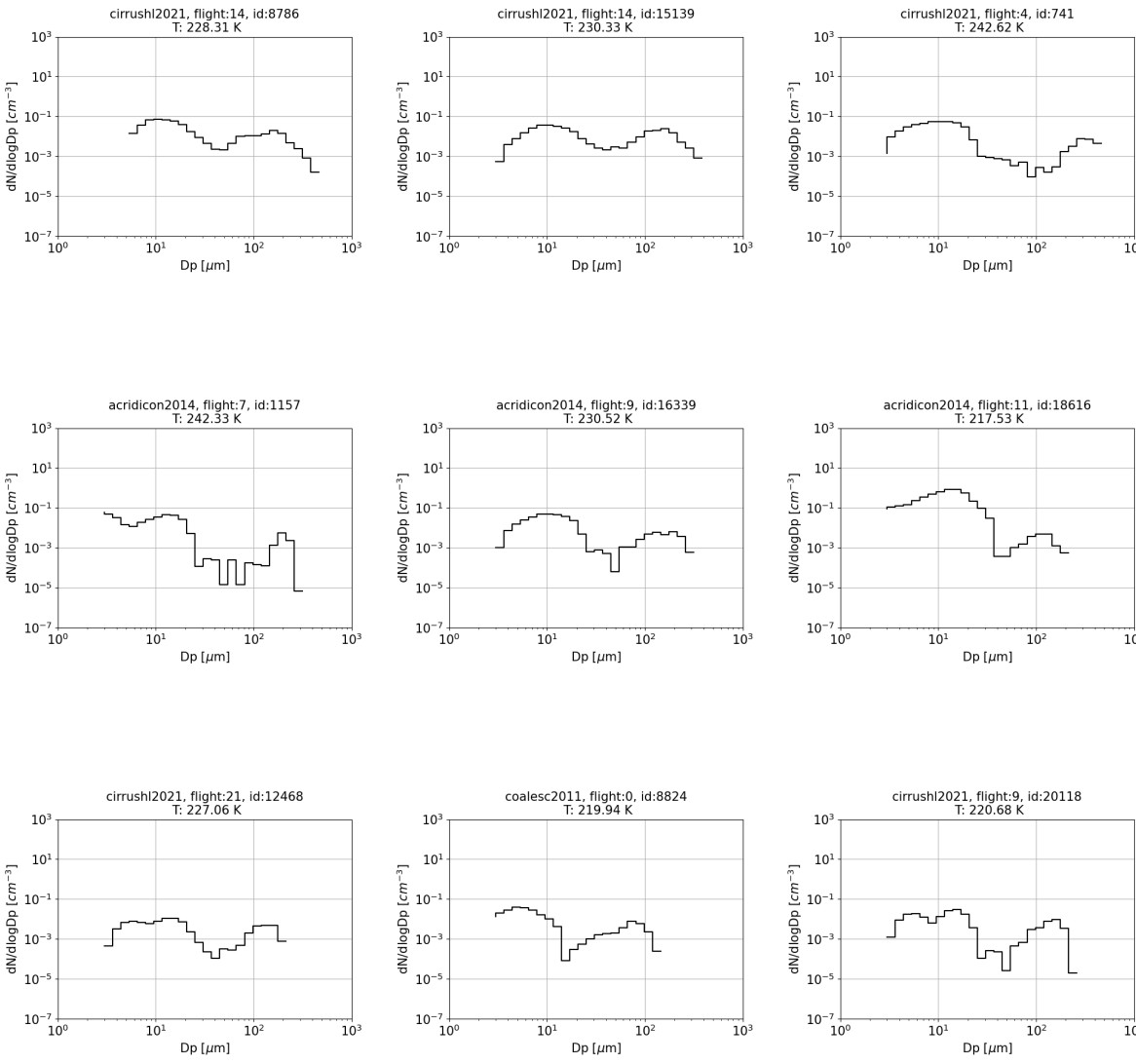

**Figure A3.** Selection of in-situ observed bimodal PSDs belonging to different airborne campaigns included in the JULIA database. In the figures the corresponding campaign, flight, number of PSD within the flight and temperature are specified. Y-axis is the concentration in $cm^{-3}$ and the x-axis is the diameter in μm.

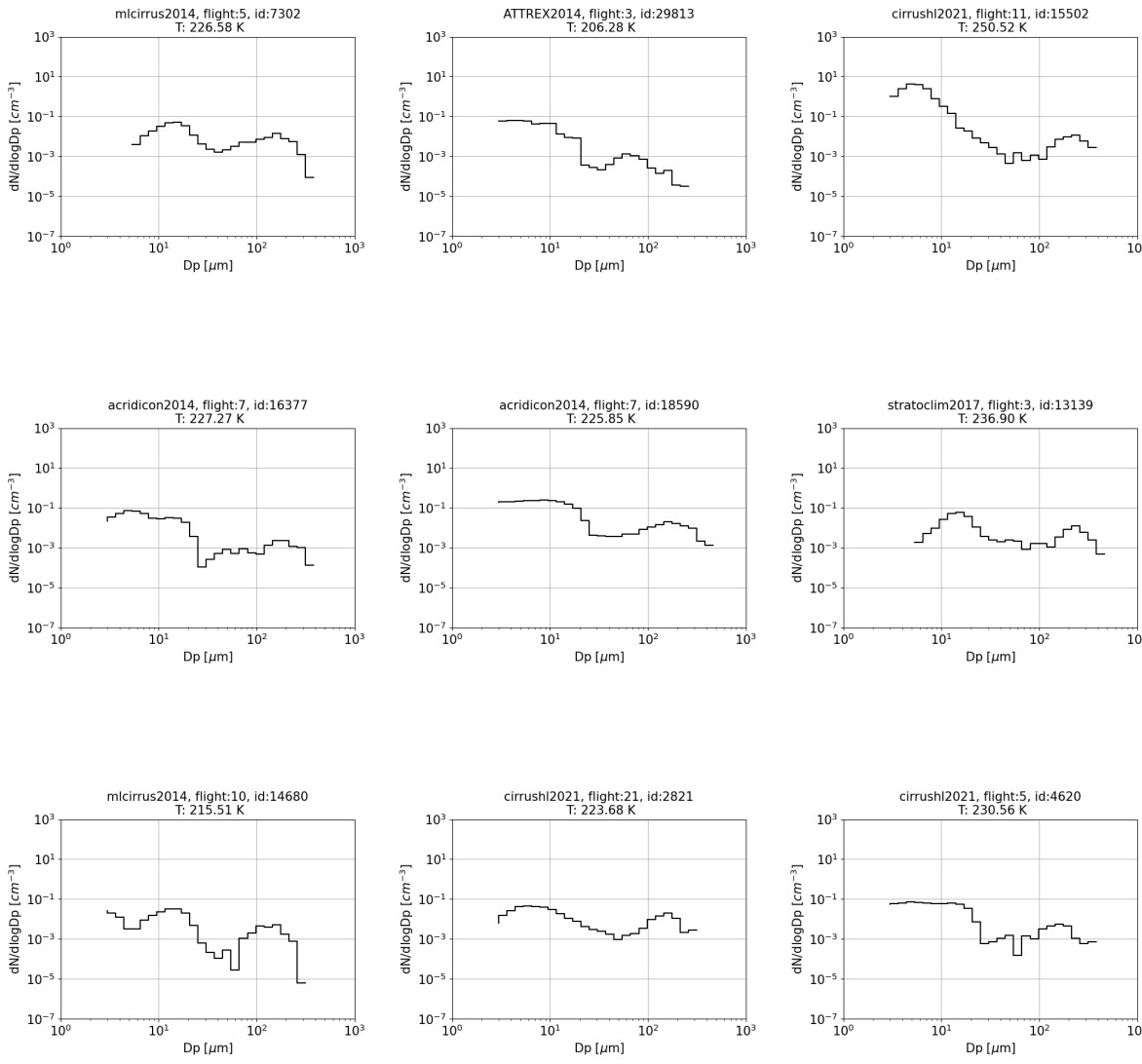

**Figure A4.** Selection of in-situ observed bimodal PSDs belonging to different airborne campaigns included in the JULIA database. In the figures the corresponding campaign, flight, number of PSD within the flight and temperature are specified. Y-axis is the concentration in $cm^{-3}$ and the x-axis is the diameter in μm.

*Code and data availability.* Please, contact the contact author

*Author contributions.* OS designed the study, IB performed the analyses, MK provided the insitu database. All co-authors contributed to the discussion of the results, improvement of the analysis and the revision of the manuscript.

*Competing interests.* At least one of the (co-)authors is a member of the editorial board of Atmospheric Chemistry and Physics.

*Acknowledgements.* The authors thank the PIRE research initiative and the Procope-Mobilität-2021 program that made possible the progress
of this study. Odran Sourdeval acknowledges support by CNES, focused on EarthCare and IASI.

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
