# Peer review of "Technical note: Bimodal Parameterizations of in situ Ice Cloud Particle Size Distributions"

_EGUsphere, 2023_

## Referee Comment (RC1)

Review of "Technical note: Bimodal Parameterizations of in situ Ice Cloud Particle Size Distributions", by Irene Garcia and coauthors, submitted to EGUSphere.

This study uses a massive set of in-situ aircraft observations collected from high latitude to equatorial ice clouds and collected in the Julia data base to investigate the size distributions of the ice and mixed-phase clouds over a wide range of conditions. Figure 1 and the cloud descriptions nicely shows the locations of the sampling, which clearly shows where the clouds were sampled. The particle probes included the CDP, FCDP, and the NIXE-CAPS, which is a CAS and grey scale CIP. The acronyms for these probes are identified in the text. Normalized size distributions, derived as a function of the melted equivalent diameter are evaluated. The interpretation of bimodality draws heavily on the data from the small particle probes. It is shown that bimodal particle size distributions (PSD) fit the observations much better than a single mode.

I have several major comments that I would like the authors to consider. A few are as follows.

1. Are the actual size distributions bimodal? Your assumed mass dimensional ($m(D)$) relationship is poorly constrained for small particles. This could affect your interpretation.
2. I've attached a figure showing PSD measured with balloon-borne ice crystal replicators, with very high resolution, no particle breakup, and unequivocal detection of small particles. There is little evidence of bimodality. It would be interesting to see if your assumed mass dimensional relationship (based on Mitchell et al., 2010) could change this result.
3. I'm very uneasy about your use of the small particle probe data. Shattering is a serious concern. The CAS is known to yield PSD that have major contributions from shattering. This issue could certainly create the bimodality you find. This issue needs to be discussed in more detail, not just in the references cited.
4. Lines. 122-124. Mass dimensional relationship. Some of your measurements are at temperatures considerably warmer than for cirrus. Is there some reason to think that you can apply the modified $m(D)$ relationship of Mitchell to the warmer temperatures?
5. Line 131 and Eq. (3). What is the advantage of using the melted equivalent diameter (from the measured PSD) versus the physical diameter. The former uses an assumed mass diameter relationship which may not be valid under certain conditions.
6. Eqs. (3) and (4). Is it valid to assume that the PSD extends from 0 to infinity, rather than a partial gamma? Does this affect the IWC?
7. Normalizing as a function of $N_{ICE}$. The value of $N_{ICE}$ is subject to considerable uncertainty and potential error.
8. Line 199. D05 and D14 use the Brown and Francis $m(D)$ relationship. How will this affect your comparison with their normalized PSD.
9. Lines 265-267. "minimize the impact of shattering effects". Down to 3 microns? This is difficult to agree with.

Minor Comments. I feel that the comments above and the few minor comments below are the ones that need to be addressed in the revised article. I'll identify more minor comments after I see the revised manuscript.

1. Line 3. based on aircraft in situ
2. 8. consists of
3. 71. What is the averaging time as that is the relevant time.
4. Eq. (2) what is the [m]
5. 149: remove studied
6. 176: "fast" to "strong"
7. 255. Parameterization

Balloon-borne Replicator Data
Multiple Ascents in Cirrus
Temperature Range: -40 to -60C

[Figure]

Maximum Diameter (microns: range, 0 to 540 microns)

---

## Author Comment (AC1)

Responses to the Referee #1 Comments on

**Technical note:**
**Bimodal Parameterizations of**
**in-situ Ice Clouds Particle Size Distributions**

by Irene Bartolomé García et al.

The authors would like to thank both anonymous referees for their comments, that helped improved the manuscript and make it more comprehensible. In the following, the comments and questions of referee #1 are addressed one by one. A reviewed version of the manuscript is attached at the end with the deleted parts in red and the new additions in blue.

**Answer to Referee#1 (RC1)**

The comments of the referee are in black,
responses by the authors in blue,
changes in the manuscript text in light blue.

Review of a Technical note: Bimodal Parameterizations of in situ Ice Cloud Particle Size Distributions, by Irene Bartolomé Garcia and coauthors, submitted to EGUSphere.

This study uses a massive set of in-situ aircraft observations collected from high latitude to equatorial ice clouds and collected in the Julia data base to investigate the size distributions of the ice and mixed-phase clouds over a wide range of conditions. Figure 1 and the cloud descriptions nicely shows the locations of the sampling, which clearly shows where the clouds were sampled. The particle probes included the CDP, FCDP, and the NIXE-CAPS, which is a CAS and grey scale CIP. The acronyms for these probes are identified in the text. Normalized size distributions, derived as a function of the melted equivalent diameter are evaluated. The interpretation of bimodality draws heavily on the data from the small particle probes. It is shown that bimodal particle size distributions (PSD) fit the observations much better than a single mode.

I have several major comments that I would like the authors to consider. A few are as follows.

1. Are the actual size distributions bimodal?

   Not every single PSD is bimodal. Our idea that a combination of two PSDs might lead to a better representation of cirrus clouds retrieved from satellite observations bases on the comparison between means of numerous measured PSDs and those retrieved from satellite observations shown by Sourdeval et al. (2018) (see Fig. 2). The observed PSDs are from a part of the campaigns which are also now included in the PSD database.
   From the two panels in Fig. 2 the bimodality of the mean observed PSDs can be seen, the more the warmer the temperatures are (especially for T > - 50°C). This is in agreement with other studies that analyzed in situ data, for example Jackson et al. (2015). In Fig. 2 it is also visible that the PSDs from the unimodal satellite retrievals deviate the more pronounced the bimodality of the observations is.

   Your assumed mass dimensional (m(D)) relationship is poorly constrained for small particles. This could affect your interpretation.

   In Afchine et al. (2018), we have tested the mass dimensional (m(D))relation versus a number of others (see Fig. 3 here) and also compared the resulting IWCs to IWCs from

total water instruments. The agreement of the IWCs was satisfactory, therefore, we are confident that the mass dimension relationship reproduces the IWC as well as possible within the given range of uncertainties.

For the smaller ice particles $\leq 100\,\mu$m), the mass dimension relation is increasingly close to spheres, which corresponds to all other m(D) relations summarized in Afchine et al. (2018) and also the newly proposed by Lawson et al. (2019), i.e. we use the best available knowledge.

2. I have attached a figure showing PSD measured with balloon-borne ice crystal replicators, with very high resolution, no particle breakup, and unequivocal detection of small particles. There is little evidence of bimodality. It would be interesting to see if your assumed mass dimensional relationship (based on Mitchell et al., 2010) could change this result.

Please see the answers to points 1. and 3., we think that the explanations given there also answer this point.

3. I'm very uneasy about your use of the small particle probe data. Shattering is a serious concern. The CAS is known to yield PSD that have major contributions from shattering. This issue could certainly create the bimodality you find. This issue needs to be discussed in more detail, not just in the references cited.

We agree that ice particle shattering played a role in earlier studies. However, the efforts made in the development and use of antishatter probe tips and particle interarrival time algorithms to minimize ice particle fragmentation have resulted in this effect no longer heavily distorting the microphysical properties of the PSDs. There are a number of publications on this issue, some of which we have cited in our manuscript. We do not feel that it is necessary to discuss the problem again in more detail, since it is 'state of the art' that shattering has been minimized as much as possible in advanced cloud probes. The following has been added to the manuscript (lines 116-120):
**As mentioned in Sect. 1, shattering of the ice particles during the measurements would increase the number of small particles and cause an artificial bimodality in the PSDs. However, as presented in the above references, major efforts were made in the development of antishatter probe tips and particle interarrival time algorithms that have resulted in a successful minimization of the shattering of ice particles. Therefore, we are confident that the bimodality present in the JULIA database is not due to distorted microphysical properties of the PSDs.**

4. Lines. 122-124. Mass dimensional relationship. Some of your measurements are at temperatures considerably warmer than for cirrus. Is there some reason to think that you can apply the modified m(D) relationship of Mitchell to the warmer emperatures?

The reason that we used the same m(D) relation at warmer temperatures is from the comparison of various m(D) relations in Afchine et al. (2018). There it can be seen that the difference between all relations is small, even when looking at relations derived for warmer temperatures. However, we are aware that the uncertainties of the derived IWCs are larger than at colder temperatures. This explanation is now included in the manuscript in Section 2.2.:
**The used $m$ - $D$ relation was compared in Afchine et al. (2018) with other $m$ - $D$**

relations from the literature and also with the measurements from total water in-
struments showing good agreement for cirrus clouds. For temperatures warmer
than the cirrus range, we are aware that the uncertainties of the derived IWC are
larger than at colder temperatures. However, we use the same $m$ - $D$, since as shown
in Afchine et al. (2018), the differences between the compared $m$ - $D$ relations is
small, even when considering those derived for warmer temperatures.

5. Line 131 and Eq. (3). What is the advantage of using the melted equivalent diameter
(from the measured PSD) versus the physical diameter. The former uses an assumed
mass diameter relationship which may not be valid under certain conditions.

The normalization method followed in our study and developed by Delanoe et al. (2005,
2014) (hereafter D05 and D14, respectively) is based on the normalization method pre-
sented by Testud et al. (2001) (hereafter T01) for raindrop spectra. We also followed this
approach. Because of the high complexity of ice particles types and shapes compared to
rain, D05 and D14 chose the equivalent melted diameter instead of the physical diameter
of the ice particles to adapt the mathematical formulation of the method of T01 to ice
PSDs. It can also be noted that m-D relations would be necessary to relate the PSD to
properties like the IWC regardless of the definition of the diameter. Please see answers to
points 1 and 2 considering the chosen mass diameter relationship and its adequacy to the
analyzed clouds.

6. Eqs. (3) and (4). Is it valid to assume that the PSD extends from 0 to infinity, rather than
a partial gamma? Does this affect the IWC?

To compute the moments of the distributions we do not use the general continuous form,
but the discrete form summing from the minimum observed diameter to the maximum
(third term in Eq. (3)).

7. Normalizing as a function of $N_{ICE}$. The value of $N_{NICE}$ is subject to considerable uncer-
tainty and potential error.

To normalize the PSD it is necessary to find a parameter to scale the size space and the
concentration space. As shown by Lee et al. (2004), a PSD can be normalized by using
combinations of moments, therefore the question is, which moments to choose. For the
normalization we are not using the total ice number concentration, which corresponds to
the zeroth moment, but a concentration metric that corresponds to the third and fourth
moment. This parameter was selected as adequate (and less uncertain than $N_{ice}$) for the
normalization process in D05 and D14 (in our manuscript Eq. (4)). These moments were
carefully selected to make normalized PSDs independent of the ice content and the mean
volume-weighted diameter.

8. Line 199. D05 and D14 use the Brown and Francis m(D) relationship. How will this
affect your comparison with their normalized PSD.

In Fig. 3, we plotted in addition to the m(D) relations shown in Afchine et al. (2018)
that of Brown and Francis (1995). It can be seen that the mass around $100\,\mu$m is higher
than those from the other m(D) relations. We suspect that in the more recent relations the
underlying measurement techniques have improved. Furthermore, in D14, it is argued
that the Brown and Francis m(D) was obtain primarily at temperatures between - $20\,°$C
and - $30\,°$C and dominated by particles between $200$ and $800\,\mu$m, so they update the study

of D05 and use m(D) relationships derived from direct IWC measurements. Therefore, we consider that the Francis and Brown m(D) might not be the most suitable one for our study since we cover colder temperatures and smaller particles.

9. Lines 265-267. 'minimize the impact of shattering effects'. Down to 3 microns? This is difficult to agree with.

   In Fig. 4, we show exemplary the mean PSD of Flight#6 of the StratoClim aircraft campaign. The inlets of the CAS probe is modified and the CIP probe is equipped with antishatter tips (see Krämer et al., 2016). In the left panel, the PSD without IAT correction to exclude shattering is shown, the right panel presents the same data set but with IAT correction applied. Comparing the two PSDs it is visible that they are nearly identical. From our analyses, ice particle shattering is generally not very frequent in cirrus clouds, because often the ice crystal sizes are not large enough to cause severe fragmentation of ice crystals. Only liquid origin cirrus sometimes carry ice crystals large enough so that an IAT correction reduces notably the number of ice crystals. In our measurements, we found only very few such events.

Minor Comments.

I feel that the comments above and the few minor comments below are the ones that need to be addressed in the revised article. I'll identify more minor comments after I see the revised manuscript.

1. Line 3. based on aircraft in situ
   Modified

2. 8. consists of
   Modified

3. 71. What is the averaging time as that is the relevant time.
   We are not averaging, we use the PSDs for every second.

4. Eq. (2) what is the [m]
   It indicates that the units of the equivalent melted diameter are meters. To avoid confusion, [m] is deleted and it is explicitly indicated in line 140.

5. 149: remove studied
   Done

6. 176: "fast" to "strong"
   We would like to keep "fast" to use the same terminology as in Krämer et al. (2016, 2020) referring to updrafts.

7. 255. Parameterization
   Modified

[Figure]

Balloon-borne Replicator Data
Multiple Ascents in Cirrus
Temperature Range: -40 to -60C

Concentration Per cm^3 Per micron Size Bin,

Maximum Diameter (microns: range, 0 to 540 microns)

Figure 1:

[Figure]

**Figure 1.** (a) Mean PSDs measured (black lines) during SPARTICUS and ATTREX, averaged per 10 °C temperature bin (from −90 to −30 °C). Black contours indicate one standard deviation around the mean. The mean and spread of one-to-one predictions by the D05 parameterization are similarly indicated in red. The total number of PSDs in each $T_c$ bin is indicated in the panel heading and the relative contributions from each campaign can be deduced from Fig. S1. Vertical plain, dashed and dotted green lines indicate $D = 5$, 25 and 100 µm, respectively. The SPARTICUS data with $T_c < −60$ °C are ignored here to avoid contaminating FCDP measurements with uncertainties arising from the first size bins of 2D-S. (b) Similar to (a) but for the ML-CIRRUS, COALESC and ACRIDICON-CHUVA campaigns.

Figure 2: Figure 1 from Sourdeval et al., 2018, ACP

[Figure]

Figure 3: Afchine et al. (2018), Figure 8 (left panel) with the m(D) relation of Brown and Francis (1995) added.

[Figure]

Figure 4: Mean PSD of Flight#6 of the StratoClim campaign. Left: without IAT correction, Right: with IAT correction.

**References**:

Afchine, A., Rolf, C., Costa, A., Spelten, N., Riese, M., Buchholz, B., Ebert, V., Heller, R., Kaufmann, S., Minikin, A., Voigt, C., Zöger, M., Smith, J., Lawson, P., Lykov, A., Khaykin, S., and Krämer, M.: Ice particle sampling from aircraft – influence of the probing position on the ice water content, Atmos. Meas. Tech., 11, 4015–4031, https://doi.org/10.5194/amt-11-4015-2018, 2018.

Delanoë, J., Protat, A., Testud, J., Bouniol, D., Heymsfield, A. J., Bansemer, A., Brown, P. R. A., and Forbes, R. M.: Statistical properties of the normalized ice particle size distribution, Journal of Geophysical Research: Atmospheres, 110, https://doi.org/https://doi.org/10.1029/2004JD005405, 2005.

Delanoë, J. M. E., Heymsfield, A. J., Protat, A., Bansemer, A., and Hogan, R. J.: Normalized particle size distribution for remote sensing application, Journal of Geophysical Research: Atmospheres, 119, 4204–4227, https://doi.org/https://doi.org/10.1002/2013JD020700, 2014.

Jackson, R. C., McFarquhar, G. M., Fridlind, A. M., and Atlas, R.: The dependence of cirrus gamma size distributions expressed as volumes in N0-λ-μ phase space and bulk cloud properties on environmental conditions: Results from the Small Ice Particles in Cirrus Experiment (SPARTICUS), J. Geophys. Res. Atmos., 120, 10,351–10,377, doi:10.1002/2015JD023492, 2015

Krämer, M., Rolf, C., Luebke, A., Afchine, A., Spelten, N., Costa, A., Meyer, J., Zöger, M., Smith, J., Herman, R. L., Buchholz, B., Ebert, V., Baumgardner, D., Borrmann, S., Klingebiel, M., and Avallone, L.: A microphysics guide to cirrus clouds – Part 1: Cirrus types, Atmos. Chem. Phys., 16, 3463–3483, https://doi.org/10.5194/acp-16-3463-2016, 2016.

Lawson, R. P., Woods, S., Jensen, E., Erfani, E., Gurganus, C., Gallagher, M., et al.: A review of ice particle shapes in cirrus formed in situ and in anvils. Journal of Geophysical Research: Atmospheres, 124, 10,049–10,090. https://doi.org/10.1029/2018JD030122, 2019.

Lee, G. W., Zawadzki, I., Szyrmer, W., Sempere-Torres, D., and Uijlenhoet R.: A general approach to double-moment normalization of drop size distributions, J. Appl. Meteor., 43, 264–281, 2004.

Sourdeval, O., Gryspeerdt, E., Krämer, M., Goren, T., Delanoë, J., Afchine, A., Hemmer, F., and Quaas, J.: Ice crystal number concentration estimates from lidar–radar satellite remote sensing – Part 1: Method and evaluation, Atmos. Chem. Phys., 18, 14327–14350, https://doi.org/10.5194/acp-18-14327-2018, 2018.

Testud, J., Oury, S., Black, R. A., Amayenc, P., and Dou, X. K.: The concept of "normalized" distribution to describe raindrop spectra: A tool for cloud physics and cloud remote sensing. J. Appl. Meteor., 40, 1118–1140, 2001.

---

## Author Comment (AC2)

Responses to the Referee #2 Comments on

**Technical note:**
**Bimodal Parameterizations of**
**in-situ Ice Clouds Particle Size Distributions**

by Irene Bartolomé García et al.

The authors would like to thank both anonymous referees for their comments, that helped improved the manuscript and make it more comprehensible. In the following, the comments and questions of referee #2 are addressed one by one. A reviewed version of the manuscript is attached at the end with the deleted parts in red and the new additions in blue.

**Answer to Referee#2 (RC2)**

The comments of the referee are in black,
responses by the authors in blue,
changes in the manuscript text in lightblue.

Review of Tecnical note: Bimodal Parameterizations of in-situ Ice Clouds Particle Size Distributions Authors: Irene Bartolomé Garcia et al.

The authors are proposing a new technic for the parameterization of ice particle size distributions with gamma normalized size distributions as in Delanoë et al 2014. But, they are using two normalized distributions, one for Diameters smaller than 50 µm and one for Diameters larger than 50 µm, instead of one for all spectrum of size of measured ice crystals. They are comparing their retrieved ice PSD with the ones of retrieved with the former methods i.e Delanoë et al., (2014 and 2005) and applied to their dataset. Globally, overall their dataset (Figure 4 and 6) the new method seems to be more accurate to retrieve small ice crystals concentration. They motivate their study, on the fact that concentrations of small ice crystals are too often neglected or not considered, impacting accuracy of retrieval methods for clouds properties. The main reason being the measurment uncertainty of small ice crystals.
This is not the first study that offers a parameterization of ice PSD with two modes (two gamma distributions cf. Field et al., 2007). However, this is the first in my knowledge that includes ice particles since 3 µm.

**Major Comments**:

1. **Bimodality:**

    (a) Are you assuming that all ice PSD in your ice clouds are bimodal?
    No, we do not assume that all PSDs are bimodal, but that bimodality can be observed in ice crystal PSDs. Futhermore, Sourdeval et al., (2018) showed using mean PSD from in situ aircraft observations compared with the retrieved mean PSD from satellite measurement that the occurence of bimodality impacts the capability of single-mode parameterization to predict the PSD shape and leads to major retrieval issues in these warmer clouds. The deviation between the mean of the PSDs is indeed clear for temperatures T > - 50°C where the bimodality is present (Fig. 1, where aggregation and possibly secondary ice production processes can occur. This is in agreement with other studies that analyzed in situ data, for example Jackson et al. (2015)

(b) Line 53: You are introducing frequencies of bimodality in the discussion, would it be consistent to divide the distribution in two modes if there is only one mode?

Since the modes of the distribution correspond one to the small particles and the other one to the large ones, even if the PSD is monomodal, it would be covered by one of the two modes or by both. In Fig.3, the rmse (root mean squared error) of the correlation between the parameterized and the observed $N_{ice}$ is compared for each of the parameterizations. It is shown, that for the monomodal ones the warmer the temperature interval, the larger the rmse is, whereas for the bimodal parameterization it remains approximately constant. Therefore, the impact of using one mode when bimodality is present (warmer temperatures) seems stronger than the use of two modes when one mode is present (colder temperatures).

(c) Hu et al 2022 have developed a method to estimate the number of modes in ice PSD. They, showed that at coldest temperature (-50° C to -40° C) ice PSD are 60% of the time one mode; except for IWC$> 1.5$ $gm^{-3}$. Why there should be bimodality?

Please, see answers to (a) and (b).

(d) In the introduction you are linking the shape of the ice PSD and the growth process. Then, it is shortly discussed in section 4.3. You are assuming that it is the difference of newly formed ice particle against sedimenting sizes. I encourage you to improve the discussion on this topic. Because, if the evolution of the size of the hydrometeors is linked to the growth rate: vapor diffusion, aggregation and riming. Then, If there is more than one growth process (without counting secondary ice production) there should be more than one mode in ice PSD !?

The following has been added to the manuscript in Sect. 4.3:

**In cirrus clouds, riming and secondary ice production play no role and aggregation is nearly negligible. These processes are of importance for mixed-phase clouds, which, as mentioned in Sect. 4.1 entail 9.8 % of the analyzed data. In Jackson et al. (2015) it was discussed that at temperatures lower than $-45°C$ the growth of the ice crystalls is likely due to depositional growth and sedimentation and aggregation are less significant. For warmer temperatures, smaller particles grow by vapor deposition and aggregation, being sedimention from above another possible source for the large particles, which together with heterogenous nucleation taking place at the same time would explain the bimodality (Zhao et al., 2018).**

(e) Then, you choose a cutting diameter of 50 $\mu$m, do you mean that the division of the growth processes such growth by vapor diffusion against growth by aggregation (or sedimentation) is here. Can you give a reference or an argument, assumption maybe, for this cutting diameter? If I observe one column of few hundreds of micron wasn't it a monocrystal of few microns in its past?

The diameter of 50 $\mu$m was first tested because it is the smallest diameter in D14, but also because it seems to agree well with the division between two modes when computing the median PSD (Fig. 2).The following has been added in Sect. 4.3:

**This cutting diameter agrees well with the division between the small and large modes when plotting the median PSD of all data (not shown).**

A diameter of 20 $\mu$m and 100 $\mu$m have been tested to see if there are major differences with the current results. 20 $\mu$m was chosen as a division between smaller particles being mainly dominated by nucleation / evaporation and larger by growth / coalescence / aggregation processess (Krämer et al., 2022). 100 $\mu$m was selected as one of the cutting diameters from Hu et al. (2022). Figure 4 compares the rmse of the correlation between retrieved $N_{ice}$ and the observed $N_{ice}$ for 20, 50 and 100 $\mu$m using the parameters specified in the manuscript (obtained using a diameter of 50 $\mu$m). There is a slight decrease for the coldest temperatures and a slight increase for the warmer ones when using a cutting diameter of 20 $\mu$m with respect 50 $\mu$m. For 100 $\mu$m there is a slight increase for all temperature intervals.

Additionally, the alpha and beta pairs have been computed using 20 $\mu$m and 100 $\mu$m. Figure 5 shows the comparison of the rmse of the correlation of parameterized and observed $N_{ice}$. It is shown that for colder temperatures the results for 20 and 50 $\mu$m are close, but the warmer the temperature, the greater the difference, being the rmse for 20 $\mu$m higher. For 100 $\mu$m, the rmse is for all temperature intervals above the rmse for 50 $\mu$m. Considering the results shown inf Fig. 4 and Fig. 5, we consider a cutting diameter of 50 $\mu$m is an adequate choice.

(f) You are citing Field et al., (2007) that also proposed a bimodal normalized parameterization, but as function of optical maximum length and effective radius. However, they did not use concentrations of small ice under 100 microns. What would be the impact by taking the concentration from 3 $\mu$m (this would be maybe to consider for a second part publication).

The parameterization by Field et al (2007), hereafter F07, is technically bimodal but only one mode was constrained with in-situ observations. The second mode, for crystals with sizes smaller than 100 $\mu$m, correspond to an exponential extrapolation. Following the Sourdeval et al (2018) study, the authors performed a similar investigation of the performance of the F07 parameterization as part of an internal evaluation for the MetOffice. Fig. 11 shows one such comparison done for the SPARTICUS campaign. It can be seen that F07 (in green; here their mid-latitude parameterization) does not perform as well as D05 for the colder temperature bins but especially that D05 and F07 perform equally poorly when bi-modality occurs. This makes the present study relevant for even parameterizations such as F07. If the study of Field et al., (2007) was updated using a database that includes ice particle size down to 3 $\mu$m (like the JULIA database used in our study), we consider that the resulting parameterization could deliver better results.

2. **Melting diameter and mass-size relations**:

(a) To retrieve the melting diameter you are using a mass-dimension relationship used in Krämer et al., (2016). In this later study, it is justified for temperature less than -38° C (235.15K) in cirrus cloud and based on former studies. Is it consistent to use it for T > 235K, knowing that few studies with direct measurment of IWC have shown an impact of the temperature on the m(D) coefficients in ice clouds.

The reason that we used the same m(D) relation at warmer temperatures is from the comparison of various m(D) relations in Afchine et al. (2018) (see Figure 8 here). There it can be seen that the difference between all relations is small, even when looking at relations derived for warmer temperatures. However, we are aware that the uncertainties of the derived IWCs are larger at warmer than at colder temperatures. This explanation is now included in the manuscript in Section 2.2.

(b) I would like to see IWC retrieved with this m(D) and original ice PSD, compared with the measured IWC available in your dataset; and also as function of temperature. Why not use, your own retrieved m(D) from the dataset you are using and see the impact on the Nice. And also with Brown and Francis as in the original version (see first review comment).

Figure 6 shows correlation plots between the retrieved IWC using the modified m(D) of Mitchell et al. (2010) together with the bimodal parameterization and the same m(D) but with the observed PSDs. Each correlation plot corresponds to a temperature range of $10\,°C$. The agreement between parameterized IWC and observed IWC is overall high, especially for temperatures between $-90°C$ and $-60°C$.

A comparison between measured IWC (with a hygrometer) and IWC from the observed PSDs and the modified m(D) of Mitchell et al. (2010) was done by Afchine et al. (2018) for two campaings (Fig. 7) showing satisfactory results. However, the comparison was made only for the colder temperature range, since the measured IWC was only available in this range. Direct measurements of IWC together with measurements of PSD are only available for one campaign, therefore it is not possible to derive our own m(D) for each single campaign as in Delanoë et al. (2014).

Regarding Brown and Francis m(D), Afchine et al. (2018) did a comparison between several m(D) relations (Fig.8). The differences were not significant, except for diameters around $100\,\mu m$ where the Brown and Francis m(D) presents a higher mass. Additionally, in Delanoë et al. (2014), it is argued that it was obtain primarily at temperatures between - 20 °C and - 30 °C and dominated by particles between 200 and $800\,\mu m$. Therefore, we consider that the Francis and Brown m(D) might not be the most suitable one for our study since we cover colder temperatures and smaller particles.

(c) Figure 4 and 5, I would consider plotting the error in percent regarding original concentrations instead of pure concentration, with a recall of your measurement uncertainties especially for smaller size. Small crystals and large ones do not have the same order of concentrations ; this is important.

Figure 9 and Fig. 10 in this response show the median percentage error of the PSD for each size bin for each parameterization. The shadowed region correspond to the range between the percentile 25 and the percentile 75. Inside the panels it is indicated the median percentage error when considering all size bins. This figures have been added to the manuscript replacing the previous figures.

(d) Figure 5 only, AS your study is questioning the retrieval of small ice concentration, I would summarize it, for small and large ice particles i.e. below and above the cutting diameter, instead of showing it as function of size bins.

In the new version of Fig. 5 (Fig. 10) for each temperature interval it is included the median error for diameters smaller and larger than $50\,\mu m$ and for the complete range of diameters.)

(e) Line 243 : I do not think that IWC and ice PSD can be dissociated. Can you be more clear on your description of the error of IWC, dimension you are using instead of log, , rate of underestimation and overestimation. It does not talk for someone who is not a specialist.

The units of the IWC are $gm^{-3}$ and the parameterized and observed IWC are compared in correlation plots similar as the ones for $N_{ice}$. Line 243 has been modified in the manuscript: **There is a slight underestimation (about 2 %) of the IWC for values between about** $1 \times 10^{-7} gm^{-3}$ **and** $1 \times 10^{-5} gm^{-3}$ **and an overestimation (about 7 %) between about** $1 \times 10^{-3} gm^{-3}$ **and** $1 gm^{-3}$.

(f) 'IWC is sensitive to large particles': it is more complicated than that. Where do you define large ice particles hundred of microns, millimeter ... The spectrum of all ice crystals goes from few microns to centimeter in some case. Then, C, S and X band radars would be enougth to retrieve IWC in cloud. For a fact, W band and Ka band do a better job i.e Delanoë et al., (2005 & 2014) which are less sensitive to very large ice crystals.

We have modified the the sentence to: **Since all parameterizations have a similar behaviour for the large particles and IWC is sensitive to large particles ($\gtrsim$ 300 μm), this result was to be expected.**

3. **Remarks on the conclusion** ;

(a) The methods of Delanoë et al., is developed for all ice clouds, while I understand that the dataset used in this study is mainly made with sampling in cirrus clouds (except for ACRIDION campaign). What about the temperature below -20° C? Can we generalize your conclusions to all ice clouds and to all range of temperature ? If yes Why ?

In our study we focus on the retrieval of ice PSDs and for temperatures lower than about -20° C. Therefore, we wouldn't generalize the results of our parameterization for warmer temperatures and we would suggest a specific study.

(b) Maybe you can recall the definition of cirrus clouds you are using, does it agree with the one in Heymsfield et al., (2017) and the AMS glossary for example?

We consider all clouds colder than -38°C to be cirrus (see Krämer et al., 2016), because at warmer temperatures clouds can also be in the mixed-phase state. This physical definition is based on the ice formation mechanism and includes in-situ origin cirrus that form directly as ice, and liquid origin cirrus, which forms at lower altitude as liquid clouds which completely galciate latest at -38°C (included now in the manuscript in Sect. 4.1). This is not entirely in line with Heymsfield et al. (2017) or the AMS glossary, however, as discussed by Heymsfield et al. (2017):
'Classifying cirrus by means of the formation mechanisms leads to cirrus types characterized by physical parameters, besides those embedded in the terminology of the WMO (1956) for all cloud types (see section 2a), which are defined based on morphology derived from observations of visual appearance.'
We are aware that these two cirrus definitions currently exist side by side and a discussion is ongoing which one should be accpeted in the future.

I suggest these references to help the discussion :

Korolev, A., Heckman, I., Wolde, M., Ackerman, A.S., Fridlind, A.M., Ladino, L.A., Lawson, R.P., Milbrandt, J., Williams, E., 2020. A new look at the environmental conditions favorable to secondary ice production. Atmospheric Chemistry and Physics 20, 1391-1429. https://doi.org/10.5194/acp-20-1391-2020.

Heymsfield, A.J., Schmitt, C., Bansemer, A., 2013. Ice Cloud Particle Size Distributions and Pressure-Dependent Terminal Velocities from In Situ Observations at Temperatures from 0° to 86°C. J. Atmos. Sci. 70, 4123â-4154. https://doi.org/10.1175/JAS-D-12-0124.1

Schmitt, C.G., Heymsfield, A.J., 2010. The Dimensional Characteristics of Ice Crystal Aggregates from Fractal Geometry. Journal of the Atmospheric Sciences 67, 1605-1616. https://doi.org/-10.1175/2009JAS3187.1

Heymsfield, A.J., Krämer, M., Luebke, A., Brown, P., Cziczo, D.J., Franklin, C., Lawson, P., Lohmann, U., McFarquhar, G., Ulanowski, Z., Tricht, K.V., 2017. Cirrus Clouds. Meteorological Monographs 58, 2.1-2.26. https://doi.org/10.1175/AMSMONOGRAPHS-D-16-0010.1

[Figure]

**Figure 1.** (a) Mean PSDs measured (black lines) during SPARTICUS and ATTREX, averaged per $10\,°\text{C}$ temperature bin (from $-90$ to $-30\,°\text{C}$). Black contours indicate one standard deviation around the mean. The mean and spread of one-to-one predictions by the D05 parameterization are similarly indicated in red. The total number of PSDs in each $T_c$ bin is indicated in the panel heading and the relative contributions from each campaign can be deduced from Fig. S1. Vertical plain, dashed and dotted green lines indicate $D = 5$, 25 and $100\,\mu\text{m}$, respectively. The SPARTICUS data with $T_c < -60\,°\text{C}$ are ignored here to avoid contaminating FCDP measurements with uncertainties arising from the first size bins of 2D-S. (b) Similar to (a) but for the ML-CIRRUS, COALESC and ACRIDICON-CHUVA campaigns.

Figure 1: Figure 1 from Sourdeval et al., 2018, ACP

[Figure]

Figure 2: Median PSD of all campaigns considering temperatures lower than 255 K.

[Figure]

Figure 3: Root mean square error (rmse) of the correlation between the parameterized ice number concentration ($N_i$) and the observed $N_i$ for several temperature intervals and for each of the parameterizations presented in the study.

[Figure]

Figure 4: Root mean square error (rmse) of the correlation between the parameterized ice number concentration ($N_i$) and the observed $N_i$ for several temperature intervals for three cutting diameters. The fitting parameters correspond to the ones specified in the manuscript.

[Figure]

Figure 5: Root mean square error (rmse) of the correlation between the parameterized ice number concentration ($N_i$) and the observed $N_i$ for several temperature intervals for three cutting diameters. The fitting parameters for the gamma function were computed for each cutting diameter.

[Figure]

Figure 6: Correlation between the parameterized IWC and the observed IWC for temperatures between -90 °C and -20 °C in intervals of 10 °C, . The parameterized IWC corresponds to the use of the bimodal parameterization. The observed IWC refers to IWC computed using the measured PSDs. The IWC was computed with units of $gm^{-3}$. Both axis correspond to the logarithm of the IWC.

[Figure]

Figure 7: Figure 11 from Afchine et al. (2018). Comparison between IWC measured with a hygrometer (y-axis) and IWC derived from a cloud spectrometer (x-axis).

[Figure]

Figure 8: Afchine et al. (2018), Figure 8 (left panel) with the m(D) relation of Brown and Francis (1995) added.

[Figure]

Figure 9: Percentage error of the parameterized PSD. The numbers inside panels (a) and (b) indicate the median error for each parameterization (D05, D14 and bimodal J2M). The shadow region in panels (c) and (d) correspond to the area between percentile 25 and percentile 75.

[Figure]

Figure 10: Percentage error of the parameterized PSD in 10 °C temperature intervals. Inside each panel the median error for diameters smaller than 50 μm, larger than 50 μm and the complete range of diameters is indicated for each parameterization (D05, D14 and bimodal J2M).

[Figure]

Figure 11: Predictions of D05 and F07 corresponding to observations during the SPARTICUS campaign.

**References**

Afchine, A., Rolf, C., Costa, A., Spelten, N., Riese, M., Buchholz, B., Ebert, V., Heller, R., Kaufmann, S., Minikin, A., Voigt, C., Zöger, M., Smith, J., Lawson, P., Lykov, A., Khaykin, S., and Krämer, M.: Ice particle sampling from aircraft – influence of the probing position on the ice water content, Atmos. Meas. Tech., 11, 4015–4031, https://doi.org/10.5194/amt-11-4015-2018, 2018.

Hu, Y., McFarquhar, G. M., Brechner, P., Wu, W., Huang, Y., Korolev, A., Protat, A., Nguyen, C., Wolde, M., Schwarzenboeck, A., Rauber, R. M., and Wang, H.: Dependence of Ice Crystal Size Distributions in High Ice Water Content Conditions on Environmental Conditions: Results from the HAIC-HIWC Cayenne Campaign. Journal of the Atmospheric Sciences, 79(12), 3103-3134. https://doi.org/10.1175/JAS-D-22-0008.1, 2022

Jackson, R. C., McFarquhar, G. M., Fridlind, A. M., and Atlas, R.: The dependence of cirrus gamma size distributions expressed as volumes in N0-λ-μ phase space and bulk cloud properties on environmental conditions: Results from the Small Ice Particles in Cirrus Experiment (SPARTICUS), J. Geophys. Res. Atmos., 120, 10,351–10,377, doi:10.1002/2015JD023492, 2015

Krämer, M., Rolf, C., Luebke, A., Afchine, A., Spelten, N., Costa, A., Meyer, J., Zöger, M., Smith, J., Herman, R. L., Buchholz, B., Ebert, V., Baumgardner, D., Borrmann, S., Klingebiel, M., and Avallone, L.: A microphysics guide to cirrus clouds – Part 1: Cirrus types, Atmos. Chem. Phys., 16, 3463–3483, https://doi.org/10.5194/acp-16-3463-2016, 2016.

Krämer, M., Spelten, N., Afchine, A., Spang, R.: Occurrence patterns of cloud particles sizes in cirrus and mixed-phase clouds, EGU22, the 24th EGU General Assembly, 23-27 May, 2022 in Vienna, Austria and Online; DOI:10.5194/egusphere-egu22-5119.
Sourdeval, O., Gryspeerdt, E., Krämer, M., Goren, T., Delanoë, J., Afchine, A., Hemmer, F., and Quaas, J.: Ice crystal number concentration estimates from lidar–radar satellite remote sensing – Part 1: Method and evaluation, Atmos. Chem. Phys., 18, 14327–14350, https://doi.org/10.5194/acp-18-14327-2018, 2018.

---

## Referee Report (RR1)

Second review of "Technical note: Bimodal Parameterizations of in situ Ice Cloud Particle Size Distributions", by Irene Garcia and coauthors, submitted to EGUSphere.

Overall, I like your responses to my first review. I have several additional comments, mostly minor,  that I would like the authors to consider in their revision of this revised article.

1. As I noted in my first review, are the actual size distributions bimodal? Your Fig. 2 shows normalized PSDs, which assumes the Kramer et al. mass dimensional relationship based on Mitchell.  Could you put in supplemental information showing PSDs from the different projects. Could the bimodality be due to shattering? Alternatively, the sample volume of the probes for the small particles is very small compared to the larger sizes, thereby making their concentration artificially large.
2. Eq. 2. The problem I see is that deriving Deq assumes a mass dimensional relationship. If D is used rather than Deq, then the PSD relationships are independent of the assumed mass dimensional relationship and are based on the measurements themselves. Could you comment on this.
3. Line 140. You mean $D_{eq}$ or D, being the physical diameter.
4. Often when cloud tops are close to or somewhat below 255K, the upper parts of the cloud are liquid or mixed-phase. I think the cutoff temperature should be perhaps 265K to completely rule out liquid water. See the article: *A global view of midlevel liquid-layer topped stratiform cloud distribution and phase partition from CALIPSO and CloudSat measurement*
5. Legend, Figure 5 d. "paremeterization" fix spelling
6. 232. "crystalls" fix spelling
7. 250. Underestimates
8. 282. Remove crystals. The larger particles might be aggregates, which wouldn't be ice crystals
9. 284. datasets

---

## Author Response (AR2)

**Technical note:**
**Bimodal Parameterizations of**
**in-situ Ice Clouds Particle Size Distributions**

by Irene Bartolomé García et al.

**Answer to second review of Referee#1 (RC1)**

The comments of the referee are in black,
responses by the authors in blue,
changes in the manuscript text in light blue.

Second review of "Technical note: Bimodal Parameterizations of in situ Ice Cloud Particle Size Distributions", by Irene Garcia and coauthors, submitted to EGUSphere.
Overall, I like your responses to my first review. I have several additional comments, mostly minor, that I would like the authors to consider in their revision of this revised article.

1. As I noted in my first review, are the actual size distributions bimodal? Your Fig. 2 shows normalized PSDs, which assumes the Krämer et al. mass dimensional relationship based on Mitchell. (a) Could you put in supplemental information showing PSDs from the different projects. (b) Could the bimodality be due to shattering? (c) Alternatively, the sample volume of the probes for the small particles is very small compared to the larger sizes, thereby making their concentration artificially large.

   **(a)** Figures 1 to 4 show a random selection of bimodal PSDs for different campaigns and different temperatures. These figures have been added as Appendix A in the manuscript. Further, PSDs in terms of frequency distributions of ice crystals for the different campaigns are shown in the paper in Figure 2. Also we like to point to the publication of Sourdeval et al. (2018), where PSDs of a number of the campaigns are shown in their Fig. 1 (please see our first answer to your points).

   **(b)** The data from the JULIA database has been carefully processed to minimize the impact of shattering as we discussed in the answers to the first review. We would like to add here a few more details about why we are confident that shattering is insignificant in the data base (see also Krämer et al., 2016; Luebke et al., 2016, Costa et al., 2017, Krämer et al. 2020): the wall of the CAS inlet entrance is "knife edged" and the inlet tips of the OAPs are modified, which greatly reduces the area susceptible for ice crystal shattering. Further, IAT (InterArrival Time) algorithms are applied to all measurements and the resulting ice concentration frequencies are carefully analyzed for shattering effects as described in Krämer et al. (2020), Appendix A2.2 (see Fig. 5): significant shattering would appear in the frequency distribution of the ice crystals as can be seen in Fig 5. In the respective graphs of the campaigns considered here and shown in the Supplementary Material of Krämer et al., 2020 (https://acp.copernicus.org/articles/20/12569/2020/acp-20-12569-2020-supplement.pdf), no bias in the frequencies is found. Therefore, we do not consider shattering to be the cause of bimodality.

**(c)** Regarding the different sample volumes, it is true that for small particles it is smaller than for the large ones (50.0 cm$^{-3}$/sec compared to 2000 - 18000 cm$^{-3}$/sec -depending on size- for an aircraft speed of 200 m/s), which can increase their concentration. However, this overestimation is reduced when having a large a amount of seconds, as described in the Appendix A2.3 of Krämer et al. (2020). In our study we have around 543000 seconds of measuremensts (see Table 2), so we consider the air volume is large enough to consider realistic concentrations, even for small particles.

2. Eq. 2. The problem I see is that deriving Deq assumes a mass dimensional relationship. If D is used rather than Deq, then the PSD relationships are independent of the assumed mass dimensional relationship and are based on the measurements themselves. Could you comment on this.

   The normalization method used in our study follows the work done by Delanoë et al. (2005), who adapted to ice particles the framework originally developed for rain by Testud et al. (2001), in the following refered as T01. The framework by T01 uses the mean volume diameter, $D_m$, which is a "volume weighted" mean diameter. Therefore, it is more convenient to use the equivalent diameter Deq and represent the ice particles with their equivalent spherical water particles, since the ice crystals are present in a wide variaty of types and shapes.

3. Line 140. You mean Deq or D, being the physical diameter.
   We mean m(D).

4. Often when cloud tops are close to or somewhat below 255K, the upper parts of the cloud are liquid or mixed-phase. I think the cutoff temperature should be perhaps 265K to completely rule out liquid water. See the article: A global view of midlevel liquid-layer topped stratiform cloud distribution and phase partition from CALIPSO and CloudSat measurement.

   Fig.6 (frorm Krämer et al. 2023, in preparation, **confidential**). shows that between 255 K and 265 K coexistence of drops and ice crystals is present in the clouds. Since we are looking at pure ice clouds that only appear below 255 K in our data set, we would like to maintain the cut off temperature at 255 K.

5. Legend, Figure 5 d. "paremeterization" fix spelling.
   Fixed.

6. 232. "crystalls" fix spelling
   Fixed.

7. 250. Underestimates
   Fixed.

8. 282. Remove crystals. The larger particles might be aggregates, which wouldn't be ice crystals.
   Fixed.

9. 284. datasets
   Fixed.

[Figure]

Figure 1: Examples of single PSDs

[Figure]

Figure 2: Examples of single PSDs

[Figure]

Figure 3: Examples of single PSDs

[Figure]

Figure 4: Examples of single PSDs

[Figure]

$N_{ice}$ $_{>3\mu m}$ diameter

**Figure A3.** Example of $N_{ice}$ measurements biased by shattering of large ice particles, visible in the frequencies of occurrence: high frequencies appear at $N_{ice}$ concentrations between 10 and $100\,cm^{-3}$ for all temperatures, which are not present in undisturbed measurements (see Fig. A2). The black lines denote the middle and maximum $N_{ice}$ lines from Krämer et al. (2009).

Figure 5: Figure adapted from Krämer et al. (2020).

[Figure]

Figure 6: Cloud particle size distributions in 10K temperature intervals, color coded by frequencies of occurence. The green lines show the median PSDs, black/white contour lines enclose 90 / 50% of the data points; Krämer et al. (2023, in preparation), **confidential**.

**Answer to second review of Referee#2 (RC2)**

The comments of the referee are in black,
responses by the authors in blue,
changes in the manuscript text in light blue.

1. Section 4.2 and 4.3: I found it difficult to understand what is exactly Julia 1M. I understand it is one mode distribution fitting, but what makes it different regards to D05 or D14? Can you clarify please.

   D05, D14 and JULIA 1M are in principle computed using the same normalization procedure adapted from the framework of Testud et al. (2001) and described in Delanoë et al. (2005). The differences are in the in situ database used for each of them, the m-D relationship and what parameters of the modified gamma function were used to minimize the cost function (i.e. to predict the in-situ data). D05 and D14 were designed to best fit optical parameters (lidar extinction and radar reflectivity) whereas JULIA 1M aims to better characterise physical parameters (IWC and Ni). Please find more details below:

   **D05**: data from the experiments CLARE98, CARL99, EUCREX, FASTEX, ARM, CEPEX and CRYSTAL FACE; m-D relationship from Brown and Francis (1995), selection of the best parameters for the modified gamma function after analysis of several combinations. Please see Delanoë et al. (2005) for more details.

   **D14**: data described in Heymsfield et al. (2013), m-D relationship used in the DARDAR products (combination of Brown and Francis (1995) for $D > 300\,\mu$m and Mitchell (1995) for hexagonal columns) and lidar extinction coefficient and radar reflectivity to minimize the cost function to chose the $\alpha$ and $\beta$ parameters of the modified gamma function. For reference, please see Delanoë et al. (2014).

   **JULIA 1M**: data from the JULIA database, modified Mitchell et al. (2010) m-D relationship described in Krämer et al. (2016) and IWC and $N_{ice}$ to minimize the cost function. The following has beed added in Sect. 4.2, lines 226-228:

   **To summarize, the parameterizations differ in the data used to compute each of them, the m-D relationship used and how the parameters of the modified gamma function were obtained.**

   Also in Sect. 4.2 we write in lines 225-226:

   **In D14 the parameters are proportional to the second and approximately sixth moment of the distribution and in our study to the zeroth moment and between the second and third moment.**

2. Section 4.3

- lines 230 : this is not well said. If in cirrus (your definition?) the aggregation processes play no role. Hence, there would not need to use two distribution (or two modes) to fit measured PSD in your dataset. But, later you say that bimodality start to play an important role for T > -60°C. I wonder about the cause of the physic processes that lead to two distributions, if it is not vapor diffusion on one side and aggregation on the other side (riming being impossible if no supercooled water)?
We added the definition we are using of cirrus at the beginning of Section 4.1: "As cirrus we understand all clouds colder than 235 K (Krämer et al., 2016). In the temperature range directly below, the clouds can also have their origin as mixed-phase clouds that have risen from below and completely glaciated latest at 235 K. This physical definition of cirrus is based on the ice formation mechanism, which is on the one hand the just mentioned complete glaciation of liquid clouds (liquid origin cirrus) and on the other hand cirrus that form directly as ice (in-situ origin cirrus)."

Regarding the causes for bimodality, we discuss in Section 4.3 that at temperatures lower than $-45\,^{\circ}C$ the growth of the ice crystals is likely due to depositional growth and sedimentation and aggregation are less significant (Jackson et al., 2015). For warmer temperatures, the ice particles also grow by vapor deposition, but sedimentation from above is a possible source for larger particles that can cause a bimodal particle spectrum (Zhao et al., 2011). Another process that can lead to bimodality is two-step ice nucleation, where there is first heterogeneous nucleation of a few ice crystals that may grow to larger sizes, followed by homogeneous nucleation of more and smaller ice crystals. However, the main reason for the bimodality of cirrus PSDs is the superposition of in-situ origin and liquid origin cirrus. Ice crystals of liquid origin are significantly larger than those of in-situ origin because they stem from lower altitudes where there is more water to allow them to grow to large sizes, especially since only very few drops out of a liquid cloud freeze so the available water vapor is deposited only among them.
This discussion has been extended between lines 235 and 242 and to avoid confusion, the following sentence in line 233 has been modified:
In cirrus clouds, riming and secondary ice production play no role and aggregation, at the coldest temperatures, is nearly negligible.

- lines 244 : Indeed, there can be many orders of magnitude between the concentration of small and large ice hydrometeors. Do large errors for large hydrometeors are less important? Maybe it needs more explanation.
No, they are not, but it is important when analysing the results to take into account that the large hydrometeors are present in lower concentrations.

- Notes about Figure 5, all parameterisations are not accurate for large hydrometeors. However, surprisingly your new parameterisation that is supposed to improve the representation of small ice crystals show more benefits in the modeling of large hdyrometeors. This is well highlighted looking median error for D > 50μm in all range of temperatures. While, median error for D < 50 are similar between D05 and Julia 2M. For the case of small ice crystals, bimodal parameterisations start to pay for the warmer temperature i.e. -30 to -20°C. Also, for temperature ranges warmer than -50°C, the Julia2M improves the representation of ice crystals from 50(20 you said) to 100 microns (the modes of large hydrometeors !?)... So adding a mode of small ice crystals do not benefit only small ice crystals but also larger !!! This is really interesting when taking into account the measurement uncertainties that are commonly admitted by the community (Baumgardner 2017) for small and large ice crystals : 100% for D < 100μm and 50% for D > 100μm. Moreover, these former parameterizations do not use concentrations for D < 50μm. Clearly, there is a need to use more than one distributions to model the concentrations of hydrometeors from pristine ice to aggregates.

  We thank the reviewer for this comment that highlights an interesting result of our study. We have added the following sentence in Section 4.3, lines 270-271:

  **As indicated by the median of the percentage error for particles smaller and larger than 50 $\mu$m, using a bimodal parameterization improves the representation of both the small and large mode, improving the large mode especially for warmer temperatures.**

  and also in Section 5, lines 301-302:

  **Adding a mode of small ice crystals do not benefit only small ice crystals but also large, despite the large measurement uncertainties associated with the large ice crystals.**

3. lines 285 : " it adjusts better to the bimodal shape of the PSDs " I would add when "it occurs".

   Added

**References**

Costa, A., Meyer, J., Afchine, A., Luebke, A., Günther, G., Dorsey, J. R., Gallagher, M. W., Ehrlich, A., Wendisch, M., Baumgardner, D., Wex, H., and Krämer, M.: Classification of Arctic, midlatitude and tropical clouds in the mixed-phase temperature regime, Atmos. Chem. Phys., 17, 12219–12238, https://doi.org/10.5194/acp-17-12219-2017, 2017.

Delanoë, J., Protat, A., Testud, J., Bouniol, D., Heymsfield, A. J., Bansemer, A., Brown, P. R. A., and Forbes, R. M.: Statistical properties of the normalized ice particle size distribution, Journal of Geophysical Research: Atmospheres, 110, https://doi.org/https://doi.org/10.1029/2004JD005405, 2005.

Delanoë, J. M. E., Heymsfield, A. J., Protat, A., Bansemer, A., and Hogan, R. J.: Normalized particle size distribution for remote sensing application, Journal of Geophysical Research: Atmospheres, 119, 4204–4227, https://doi.org/https://doi.org/10.1002/2013JD020700, 2014.

Heymsfield, A. J., Schmitt, C. G., and Bansemer, A. R.: Ice cloud particle size distributions and pressure-dependent terminal velocities from in situ observations at temperatures from 0° to -86°C. Journal Of The Atmospheric Sciences, 70, 4123-4154. doi:10.1175/JAS-D-12-0124.1, 2013.

Jackson, R. C., McFarquhar, G. M., Fridlind, A. M., and Atlas, R.: The dependence of cirrus gamma size distributions expressed as volumes in N0-λ-µ phase space and bulk cloud properties on environmental conditions: Results from the Small Ice Particles in Cirrus Experiment (SPARTICUS), J. Geophys. Res. Atmos., 120, 10,351–10,377, doi:10.1002/2015JD023492, 2015.

Krämer, M., Rolf, C., Luebke, A., Afchine, A., Spelten, N., Costa, A., Meyer, J., Zöger, M., Smith, J., Herman, R. L., Buchholz, B., Ebert, V., Baumgardner, D., Borrmann, S., Klingebiel, M., and Avallone, L.: A microphysics guide to cirrus clouds – Part 1: Cirrus types, Atmos. Chem. Phys., 16, 3463–3483, https://doi.org/10.5194/acp-16-3463-2016, 2016.

Krämer, M., Rolf, C., Spelten, N., Afchine, A., Fahey, D., Jensen, E., Khaykin, S., Kuhn, T., Lawson, P., Lykov, A., Pan, L. L., Riese, M., Rollins, A., Stroh, F., Thornberry, T., Wolf, V., Woods, S., Spichtinger, P., Quaas, J., and Sourdeval, O.: A microphysics guide to cirrus – Part 2: Climatologies of clouds and humidity from observations, Atmospheric Chemistry and Physics (highlight article), 20, 12 569–12 608, https://doi.org/10.5194/acp-20-12569-2020, 2020.

Luebke, A. E., Afchine, A., Costa, A., Grooß, J.-U., Meyer, J., Rolf, C., Spelten, N., Avallone, L. M., Baumgardner, D., and Krämer, M.: The origin of midlatitude ice clouds and the resulting influence on their microphysical properties, Atmos. Chem. Phys., 16, 5793–5809, https://doi.org/10.5194/acp-16-5793-2016, 2016.

Testud, J., Oury, S., Black, R. A., Amayenc, P., and Dou, X. K.: The concept of "normalized" distribution to describe raindrop spectra: A tool for cloud physics and cloud remote sensing. J. Appl. Meteor., 40, 1118–1140, 2001.

Zhao, Y., Mace, G. G., and Comstock, J. M.: The Occurrence of Particle Size Distribution Bimodality in Midlatitude Cirrus as Inferred from Ground-Based Remote Sensing Data. Journal of the Atmospheric Sciences, 68(6), 1162-1177. https://doi.org/10.1175/2010JAS3354.1, 2011.

---

## Author Response (AR3)

**Technical note:**
**Bimodal Parameterizations of**
**in-situ Ice Clouds Particle Size Distributions**
**by Irene Bartolomé García et al.**

**Answer to the Editor**
The comments of the Editor are in black,
responses by the authors in blue,
changes in the manuscript text in light blue

**Minor revision**

06 Dec 2023
Editor decision: Publish subject to technical corrections
by Barbara Ervens
**Public justification (visible to the public if the article is accepted and published)**:
Dear Authors,
many thanks for addressing the remaining referee comments. I am happy to accept your paper for publication in ACP.
Prior to uploading your files for paper production, please fix the minor/technical issues as listed below.
Sincerely,
Barbara Ervens
=========================
**Line numbers refer to the manuscript version without annotations.**
**l. 63: analyzes**
Done

**l. 71: Please define IWC here (unless I missed it before)**
Done

**l. 86: consisted of**
Done

**Table1: State at least in the caption what 'ranges' refer to (e.g., '...of ice particle diameters') so that the table is more self-explanatory**
Done

**l. 128, 131, 202, 202 (and possibly other instances): 'warm temperature' is scientifically not fully correct:**
**temperature denotes a value – which can be high or low; warm/cold describes an intensive property of matter (e.g. gases, ice, water). Thus, high temperature leads to warming.**
Done

**l. 131: can you specify the temperature range for which it is valid?**
The temperature range covered in the m-D relationships used in the comparison in Afchine et al. (2018) are:
- Heymsfield et al. (2010): -60 ˚C < T < 0 ˚C
- Mitchell et al. (2010):  -60 ˚C < T < -20 ˚C
- Cotton et al. (2013): -60 ˚C < T < -20 ˚C
- Erfani and Mitchell (2016): -65 ˚C < T < -20 ˚C

The following line has been added in lines 127-128: …with other m-D relations from the literature (covering, depending on the m-D relation, temperatures between -65 ˚C and 0˚C)

**l. 135: should it be 'using' and 'fitting' to logically continue the list of steps starting with 'computing'?**
Done

**l. 140: Is it relevant that Dm is in units of meters? This equation is generally valid, independently of units, provided consistent units, e.g. mass in kg and density in kg/m^3. I understand that the coefficients alpha, beta might have been derived using SI units (kg, m, …) but this could be generally stated around Eq.-6.**
It should be Deq instead of Dm. We specified the used units for consistency with the description of the method in D14.

**l. 160 ff: The equations should be numbered with separate numbers, i.e. 7, 8, 9. Make sure to refer to them accordingly in the text (e.g. l. 207)**
Done

**l. 175/6: The new sentence does not read well. May be better something like: In the temperature range just below 235 K, the clouds may originate as mixed-phase clouds ascending from lower altitudes, undergoing complete glaciation at ≥ 235 K.**
Done

**l. 181: Either 'in a temperature range of' or 'at temperatures'**
Changed to 'at temperatures'

**Table 2: Please improve the table caption so that the table is more self-explanatory**
Done

**l. 191: Either 'another indicator of cirri that have… '(https://www.merriam-webster.com/dictionary/cirrus) or 'another indicator of cirrus clouds…'**
Changed to cirrus clouds

**l. 206: define (remove 'd')**
Done

**l. 229: each of them**
Done

**Figure 3, caption: What do you mean by 'left triangle'? The symbol for the new parameterization (JULIA 1 M) looks to me like a circle.**
Changed to 'black circle'

**l. 233: ...do not play any role**
Done

**l. 235/6: replace 'and' by 'whereas' to avoid ambiguity:**
**due to depositional growth WHEREAS sedimentation and aggregation are less significant.**
Done

**l. 238/9: The structure of the new sentence does not seem right (verb missing?). Maybe better:**
**Initially, the few heterogeneously nucleated ice crystals may grow to larger sizes, followed by ...**
Done

**l. 254: Better: From Fig. 4a and 4c... (to avoid confusion as you refer to Fig 2 in the previous sentence... which doesn't even have a, b, c...).**
Done

**l. 256, 260: panel b, d should be Fig 4b, 4d etc (see previous comment)**
Done

**l. 271: small and large modes (add 's')**
Done

**l. 281: 'subset' cannot be used as a verb. Better:**
**Binning this data into 10-K temperature intervals between -90C and -60 C...**
Done

**l. 284/5: Please clarify: (i) what do you mean by 'temperature ranges' (also Fig 5 caption)? Temperature intervals (or 'bins')?**
**(ii) intervals of 10 degree C and 10 K are the same. What exactly did you compare here?**
We mean temperature intervals. In the caption of Fig. 5 K has been replaced with ˚C.

**l. 290: do you mean indeed 'when' or rather 'if' (implying that it is not always the case)?**
We mean when. Not all PSDs are bimodal, but when there are bimodal PSDs, having the two modes fits better the observations than having only one mode.

**Figure 4, caption: Is there a word missing at the end? Over the complete size RANGE?**
Added the word "range"

**l. 301/2: The new sentence does not read well. Please improve. My suggestion (check whether reflects the intended meaning!):**
**Considering a second mode improves the PSD prediction of both small and large ice crystals despite the large measurement uncertainties associated with the latter.**
Done